# Spike-based dynamic computing with asynchronous sensing-computing neuromorphic chip

Man Yao ®[1,12], Ole Richter ®[2,12], Guangshe Zhao[3,12], Ning Qiao[2,4,12], Yannan Xing[4], Dingheng Wang[5], Tianxiang Hu[1], Wei Fang ®[6,7], Tugba Demirci[2], Michele De Marchi[2], Lei Deng ®[8], Tianyi Yan ®[9], Carsten Nielsen[2,10], Sadique Sheik[2], Chenxi Wu[2,10], Yonghong Tian ®[6,7], Bo Xu[1] & Guoqi Li[1,11] ✉

By mimicking the neurons and synapses of the human brain and employing spiking neural networks on neuromorphic chips, neuromorphic computing offers a promising energy-efficient machine intelligence. How to borrow high-level brain dynamic mechanisms to help neuromorphic computing achieve energy advantages is a fundamental issue. This work presents an application-oriented algorithm-software-hardware co-designed neuromorphic system for this issue. First, we design and fabricate an asynchronous chip called "Speck", a sensing-computing neuromorphic system on chip. With the low processor resting power of 0.42mW, Speck can satisfy the hardware requirements of dynamic computing: no-input consumes no energy. Second, we uncover the "dynamic imbalance" in spiking neural networks and develop an attention-based framework for achieving the algorithmic requirements of dynamic computing: varied inputs consume energy with large variance. Together, we demonstrate a neuromorphic system with real-time power as low as 0.70mW. This work exhibits the promising potentials of neuromorphic computing with its asynchronous event-driven, sparse, and dynamic nature.

Resource and energy constraints are the major restrictions to deploying traditional AI methods, especially in real-world edge platforms. A promising solution with an attractive low-power feature is neuromorphic computing, which is partially inspired by the human brain that runs even more complex and larger neural networks with a total energy need of just 20 W[1–4]. By abstracting the computations in the human brain at the neuron and synapse level, existing neuromorphic platforms, such as the classic BrainScales[5], SpiNNaker[6], Neurogrid[7], TrueNorth[8], and the most recent Darwin[9], Loihi[10], Tianjic[11],

have demonstrated impressive energy efficiency via spike-based communication and computing. However, whether this level of abstraction[2,12,13] is the most suitable approach for emulating the efficient computation of the brain, and the role that high-level stereo brain mechanisms can play in neuromorphic chips, are challenges that must be addressed at this stage".

An important function of the human brain is the ability to dynamically allocate its resources according to the required demand, which is what we call "dynamic computing" due to the attention

[1]Institute of Automation, Chinese Academy of Sciences, Beijing, China. [2]SynSense AG Corporation, Zurich, Switzerland. [3]School of Automation Science and Engineering, Xi'an Jiaotong University, Xi'an, Shaanxi, China. [4]SynSense Corporation, Chengdu, Sichuan, China. [5]Northwest Institute of Mechanical & Electrical Engineering, Xianyang, Shaanxi, China. [6]School of Computer Science, Peking University, Beijing, China. [7]Peng Cheng Laboratory, Shenzhen, Guangdong, China. [8]Center for Brain-Inspired Computing, Department of Precision Instrument, Tsinghua University, Beijing, China. [9]School of Life Science, Beijing Institute of Technology, Beijing, China. [10]Institute of Neuroinformatics, University of Zurich and ETH Zurich, Zurich, Switzerland. [11]Key Laboratory of Brain Cognition and Brain-inspired Intelligence Technology, Beijing, China. [12]These authors contributed equally: Man Yao, Ole Richter, Guangshe Zhao, Ning Qiao. ✉e-mail: guoqi.li@ia.ac.cn

mechanism[14,15]. Salient stimuli tend to receive greater attention, primarily manifested in the heightened spiking activity of brain regions or neurons associated with the stimulus[16]. Incorporating the high-level dynamic computing nature of the human brain into machine intelligence is very challenging. Specifically, dynamic computing encompasses two connotations: energy consumption is minimal when there is no input, while it significantly varies with input changes. With these understandings as the anchor, we present an application-oriented algorithm-software-hardware co-designed neuromorphic system to investigate the dynamic and sparse computing of spike-based machine intelligence in our newly designed and fabricated neuromorphic chip.

To achieve the hardware requirement of no-input consumes no running energy, we design and fabricate the "Speck" (Fig. S1) with the size of 6.1 mm × 4.9 mm, a spike-based and fully asynchronous neuromorphic chip with low processor resting power (only **0.42 mW**). The fully asynchronous architecture of Speck, which renders computing capacity solely dependent on input data, constitutes the key factor behind its persistent "always-on" profile. In this paradigm, the neuromorphic chip no longer needs the global or local clock signal, which efficiently prevents the redundant power consumed by clock empty flips. In other words, the asynchronous design can be understood as the most extreme form of fine granular clock gating for every component in the processing pipeline while being instantly available, requiring no wake-up procedures[17]. Meanwhile, by integrating the DVS[18–20] as the "eye" of the chip, Speck becomes the sensing-computing neuromorphic System on Chip (SoC). The DVS asynchronously and sparsely generates a stream of events (binary spikes with addresses) when the brightness of the visual scene changes. The processor in Speck only operates when receiving incoming events, leveraging its hardware circuit design to enable asynchronous event-driven distributed convolution processing of spike trains. Remarkably, the entire system processes a single spike with an ultra-low latency of only **3.36** μs. This collaborative philosophy between neuromorphic hardware and applications perfectly encapsulates the essence of dynamic computing. It empowers Speck with distinct advantages in scenarios with stringent power and latency requirements, such as mobile devices and the Internet of Things.

In the context of spike-based computing, it is commonly believed that computation is triggered exclusively by input spikes. With each input, only a subset of the network becomes activated, resulting in the activation of multiple sets of spiking neurons. Therefore, it is natural to believe that varied inputs consume different energy in SNNs. However, we uncover a phenomenon called "dynamic imbalance" that commonly exists but has been ignored for a long time in SNNs, i.e., although spiking neurons are selectively and sparsely activated, spiking networks respond similarly to different inputs. Specifically, we observe that spiking firing rates in vanilla SNNs at each timestep are very similar, which indicates that the scales of the activated sub-networks are similar for diverse input. It implies the connotations of dynamic computing referring to "varied inputs consume energy with large variance" usually does not hold in SNNs. Consequently, the dynamic computing advantage that neuromorphic system naturally have is undermined. To address this issue, we design an attention-based dynamic framework, which can assist SNNs in regulating spiking responses according to the importance of input discriminatively.

To efficiently deploy algorithms/models for various dynamic vision applications, Speck provides a complete software toolchain, including data management, model simulation, host management, etc. This enables us to demonstrate the attractive features of the proposed neuromorphic system in accuracy, energy cost, and latency. To this end, dynamic SNNs are evaluated on four demanding event-based action recognition benchmarks. Extensive experiments show that the attention nature of the brain, data-dependent dynamic processing currently underappreciated in SNNs, can confer sparser firing and

better performance to SNNs concurrently. By deploying the dynamic SNNs to Speck, we demonstrate a high-accuracy neuromorphic system with real-time power as low as **0.70 mW** and ultra-low latency of less than **0.1 ms** on a single sample in public datasets. The practice in this work demonstrates the power of the neuromorphic chip in dynamic computing, expands a creative path for the development of neuromorphic computing, and pushes neuromorphic computing a big step toward real-world applications.

## Results

Dynamic computing is an emerging topic in deep learning[21]. Dynamic neural networks can adapt their computational graphs (activated structures) to the input at the inference stage, thus holding more attractive properties than static ones, such as performance, computational efficiency, etc. From the perspective of dynamic computing, neuromorphic and traditional AI systems are two completely different paradigms. Neurons in SNNs communicate through spike (0-nothing or 1-spike) trains, while neurons in traditional Artificial Neural Networks (ANNs) exchange information using continuous values (Fig. 1a). Consequently, spike-based neuromorphic computing naturally has a dynamic computational graph, with often only a small percentage of the overall spiking neurons being active at any moment and the rest being idle. In contrast, traditional computing, such as ANNs working on GPUs, is controlled by static computational graphs. Even if all inputs or activations of ANNs are zeros, the network must perform all operations (Fig. 1b). To realize dynamic computing in traditional AI, researchers have designed various dynamic algorithms[21–23].

However, the energy-efficiency advantages of dynamic computing in real AI systems remain theoretical for now. Hardware is the first and most challenging obstacle to overcome. The power required to run an AI system usually consists of two components, resting and running power. The former is determined by the hardware design, while the latter depends on the model deployed when the hardware is fixed (Fig. 2a). The great majority of hardware, whether neuromorphic or traditional, employs a significant amount of energy even when no computing is being done. Suppose the ratio of resting power to the overall power is too high, it is illogical to expect the total power can be lowered by dynamic algorithm design that decreases the running power only (Fig. 2b, c).

Even considering only the running power, gaining the benefits of advanced dynamic algorithms on hardware is challenging. For traditional AI systems, there will inevitably be a gap between dynamic algorithms and hardware practical efficiency when running dynamic sparse patterns (e.g., indexing, weight-coping) on dense computing hardware[21]. For neuromorphic AI systems, they are theoretically born with a form of dynamic computing that only activates a subset of spiking neurons for any input. However, an overlooked fact is that vanilla SNNs don't possess the function of dynamic computing (i.e., dynamic imbalance), which requires discriminative responses to 'easy' or 'hard' inputs. Specifically, we observe that the Network Spiking Firing Rate (NSFR) of SNN is almost constant throughout timesteps (Fig. 1c). We argue that SNNs' spatio-temporal invariance[24,25], which exploits the same weights for each location across all timesteps, is the underlying cause of this phenomenon (Fig. S2). This sharing weakens the potential energy benefit of dynamic computing (Fig. S3), despite increasing the parameter utilization efficiency. In this work, we present a neuromorphic system that practices and demonstrates the power of dynamic computing.

### An always-on sensing-computing neuromorphic SoC − Speck

Most neuromorphic hardware design begins from the bottom of the compute stack, i.e., the materials and devices, and it is then the developer's responsibility to map corresponding algorithms and applications onto the hardware[2]. Contrarily, the design ethos of Speck is that neuromorphic hardware deployed at the edge may be tailored

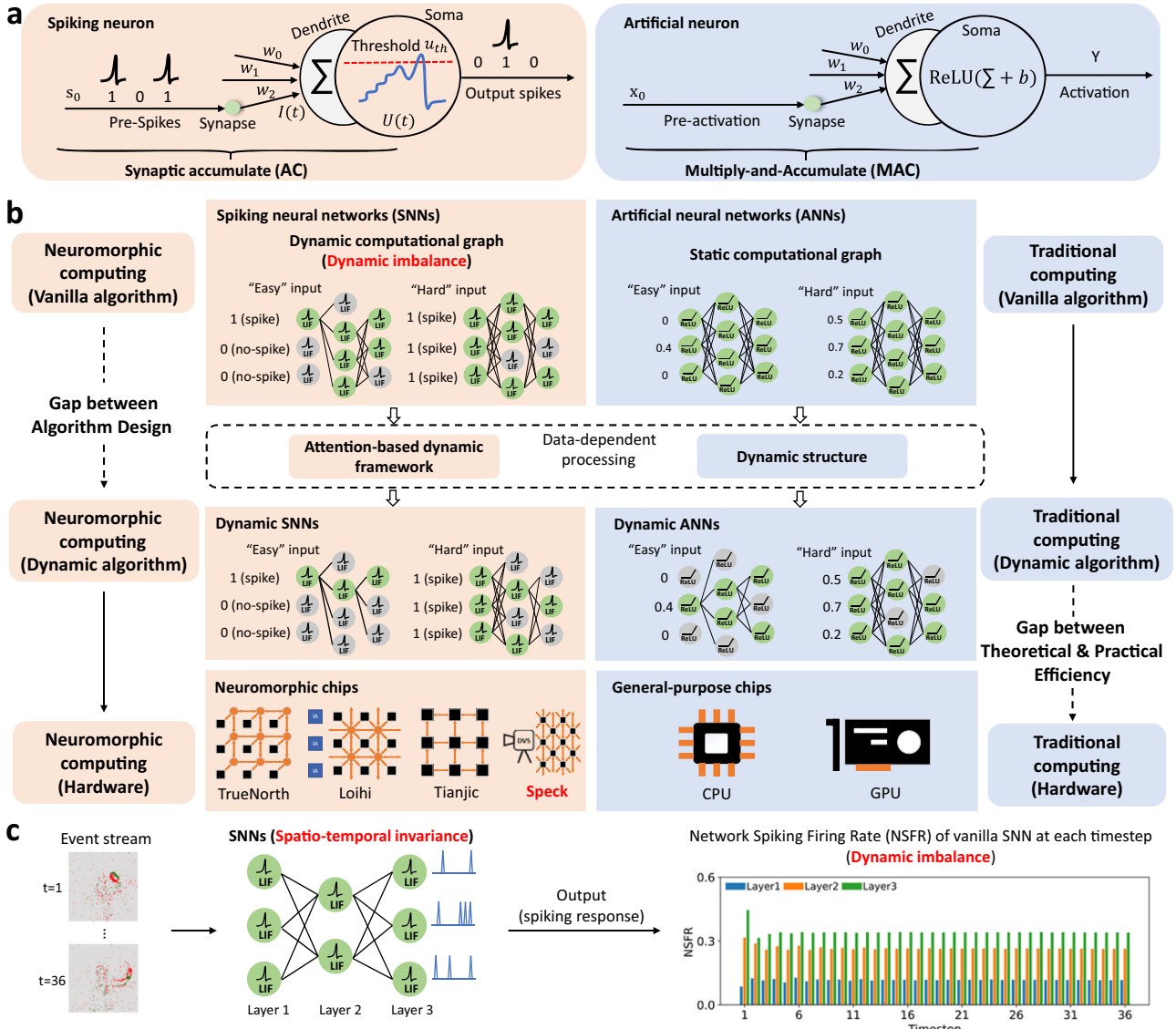

**Fig. 1 | Neuromorphic computing vs. traditional computing from the view of dynamic computing. a** Spiking neuron vs. Artificial neuron. Left: spiking neurons communicate through spike trains coded in binary spikes, and the major operation is synaptic Accumulation (AC) between weights. Right: neurons in ANNs communicate using activations coded in analog values, and Multiply-and-Accumulate (MAC) of inputs and weights is the major operation. **b** From a dynamic computing perspective, we compare neuromorphic and traditional computing in three aspects: vanilla algorithm (top), dynamic algorithm (middle), and hardware (bottom). In traditional computing (right part), vanilla algorithms generally hold a static, fixed computational graph manner. Although some neurons have activation values of zero, all zero-based MAC operations must be performed. By adapting the structures of static models to different inputs, dynamic ANNs can lead to notable advantages in accuracy and computational efficiency. However, traditional computing hardware is mostly optimized for static models and not friendly to dynamic networks, and there is a gap between the theoretical and practical efficiency of dynamic ANNs[21]. In neuromorphic computing (left part), SNNs are born with dynamic computational graphs, and neuromorphic hardware is naturally suitable for SNNs. However, we observed the dynamic imbalance in SNNs, which respond similarly to diverse inputs. **c** Dynamic imbalance. SNNs satisfy dynamic activation forms but are not good at dynamic functions, i.e. responding discriminatively. Spatio-temporal invariance (Figs. S2, S3) is the fundamental assumption of SNNs because they share parameters at different timesteps. Consequently, LSFRs (definition is given in Part "Details of algorithm evaluation") at each timestep is similar, which indicates that the scales of the activated sub-networks of SNNs are similar for diverse input.

to operate with one particular application and have a focus point (Fig. 2e), such as low power or latency. Then, the shining unique benefits of neuromorphic computing might be able to speed up the development of the technology.

As an integrated sensing-computing neuromorphic SoC, Speck (Fig. 2d, Fig. S1, Table S1) combines a 128 × 128-pixel DVS with an asynchronous spike-based neuromorphic AI chip. Speck contains 328 k spiking neurons, with an integration level of 11,000 neurons per square millimeter. Its processing pipeline is built with asynchronous digital logic[26], thus enabling always-on low resting power consumption

(Fig. S6, Table S2) and optimum latency. Specifically, asynchronous logic is built using a cascade of asynchronous circuit blocks that communicate via a request/acknowledge protocol when transferring data, without requiring a global clock. As soon as the data is accessible at its input port, each block independently calculates its output value. Thus, no running power from logic gates is employed when the asynchronous pipeline is idle.

Nonetheless, asynchronous logic is far less used than synchronous digital logic in Von-Neumann processor architecture implementations. The primary cause is that the design and implementation

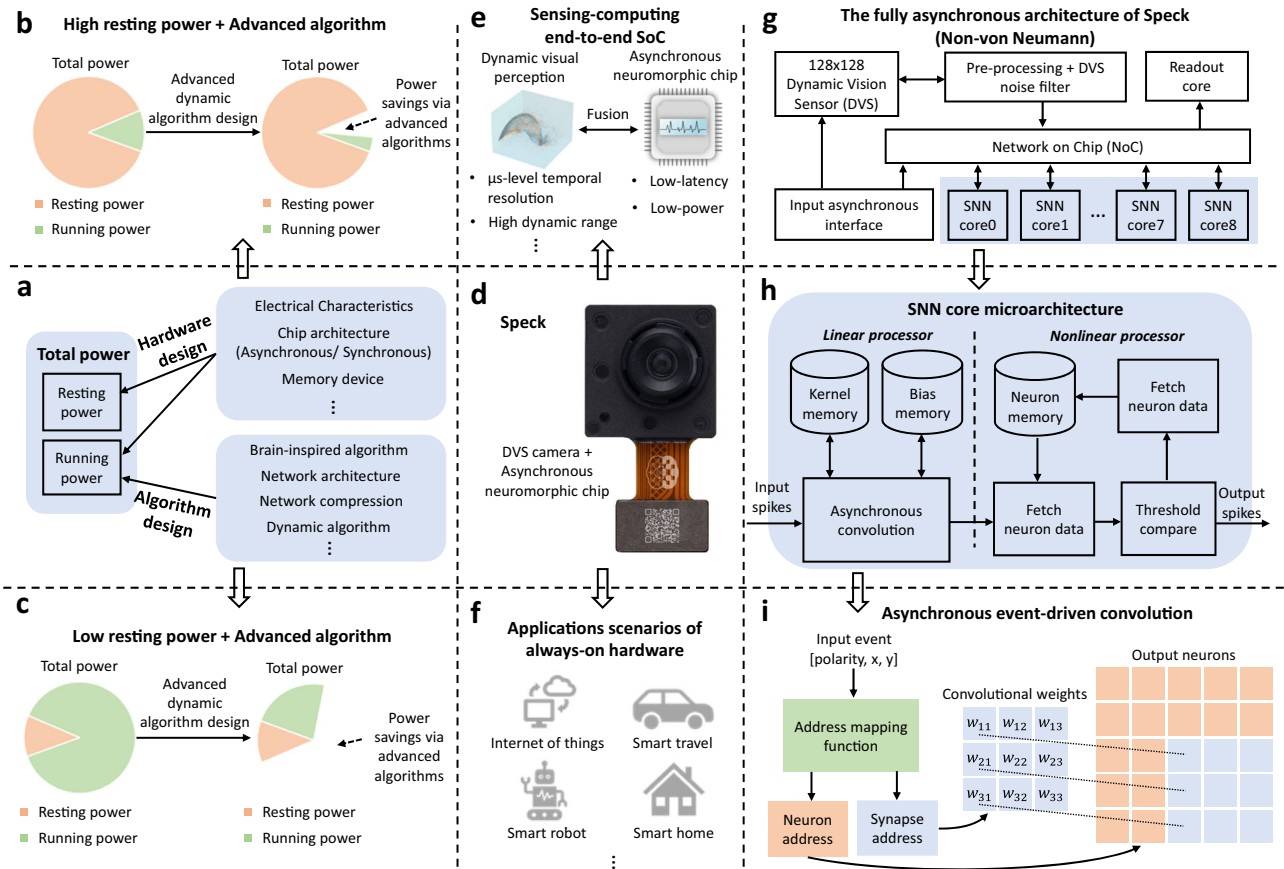

**Fig. 2 | The design details of Speck. a** The power composition of AI systems. **b** The case of high resting power. When the resting power is too high, the gain brought by the advanced algorithm design is hard to lower the total power effectively. **c** The case of low resting power. Low resting power helps unleash the power of advanced algorithm design. **d** Speck physical display. **e** Speck is a sensing-computing end-to-end SoC that integrates DVS and asynchronous neuromorphic chip. **f** Typical application scenarios of always-on Speck. **g** The fully asynchronous architecture of Speck. The DVS events come from the on-chip sensor. After an asynchronous event pre-processing core, events can be routed to SNN cores for processing. In Fig. S1,

we give the layout of Speck. **h** The SNN core microarchitecture (more details in Fig. S5). Each SNN core can be simply considered as a spiking convolution layer with an integrated pooling layer. When a spike (event) is received at the input of the core, a fully asynchronous convolution operation is performed to calculate all required neuron updates caused by the received input spike. **i** Asynchronous event-driven convolution. Based on the address of the input event or spike, the address mapping function outputs the address of the neuron and synapse that need to perform synaptic operations (more details in Fig. S4).

of asynchronous circuits are more complicated. To tackle this challenge, we optimized the overall sensing to computing strategy with event streams in the dynamic visual scene, and thus we can take full advantage of the spike nature of the data and the modular architecture of the chip to perform flexible, efficient event-driven computing. Specifically, Speck comprises a central event router that can be configured to route events from any to any of the 9-SNN cores, where every core can work independently and asynchronously (Fig. 2g). Consequently, the design effort is limited to a single SNN core (Fig. 2h) but can be scaled to design larger hardware. Moreover, asynchronous event-driven convolution is also one of the core designs to improve computational efficiency (Fig. 2i, Fig. S4).

### Attention-based dynamic framework for SNNs

It has long been known that the brain's dynamic visual responses are associated with both external visual stimuli and the brain's internal attention mechanisms. Some brain attention-based stimulus-related functional modulations are easily described[15,16]. Attention is a limited resource, and brains typically respond more favorably to a preferred stimulus while inhibiting a non-preferred stimulus. These functional processes are mainly reflected in the activeness of the spikes in the brain. On the other hand, as a powerful and complicated brain mechanism, attention stereoscopically acts on diverse neural levels. In

Fig. 3a, we roughly divide neural correlates of attention into four structural levels. These prior understandings of visual attention in neuroscience provide us with a lot of inspiration for designing dynamic SNNs.

Our goal is to mitigate the dynamic imbalance of SNNs so that they can react to varied inputs more discriminatively. In the human brain, attention controls the neuronal activity to perform this goal. Inspired by this, the basic idea of the proposed dynamic framework is to modify the spiking response by optimizing the membrane potential because spiking neurons determine whether to fire based on whether the membrane potential exceeds a threshold. In our previous work[25,27–29], we implemented the attention through additional plug-and-play modules, including independent or coupled three dimensions of temporal, channel, and spatial, to learn "when", "what" and "where" to focus on. These attention modules first capture global information of different dimensions and then use them to model the relative importance between different input moments, channels, or locations. The attention modules finally output the attention scores (a value between 0 and 1), which are exploited to optimize the membrane potential.

To facilitate the deployment of attention SNNs on neuromorphic chips, we summarize existing methods[25,27–29] into a general attention-based dynamic framework. Figure 3b, c show the LIF-SNN layer and

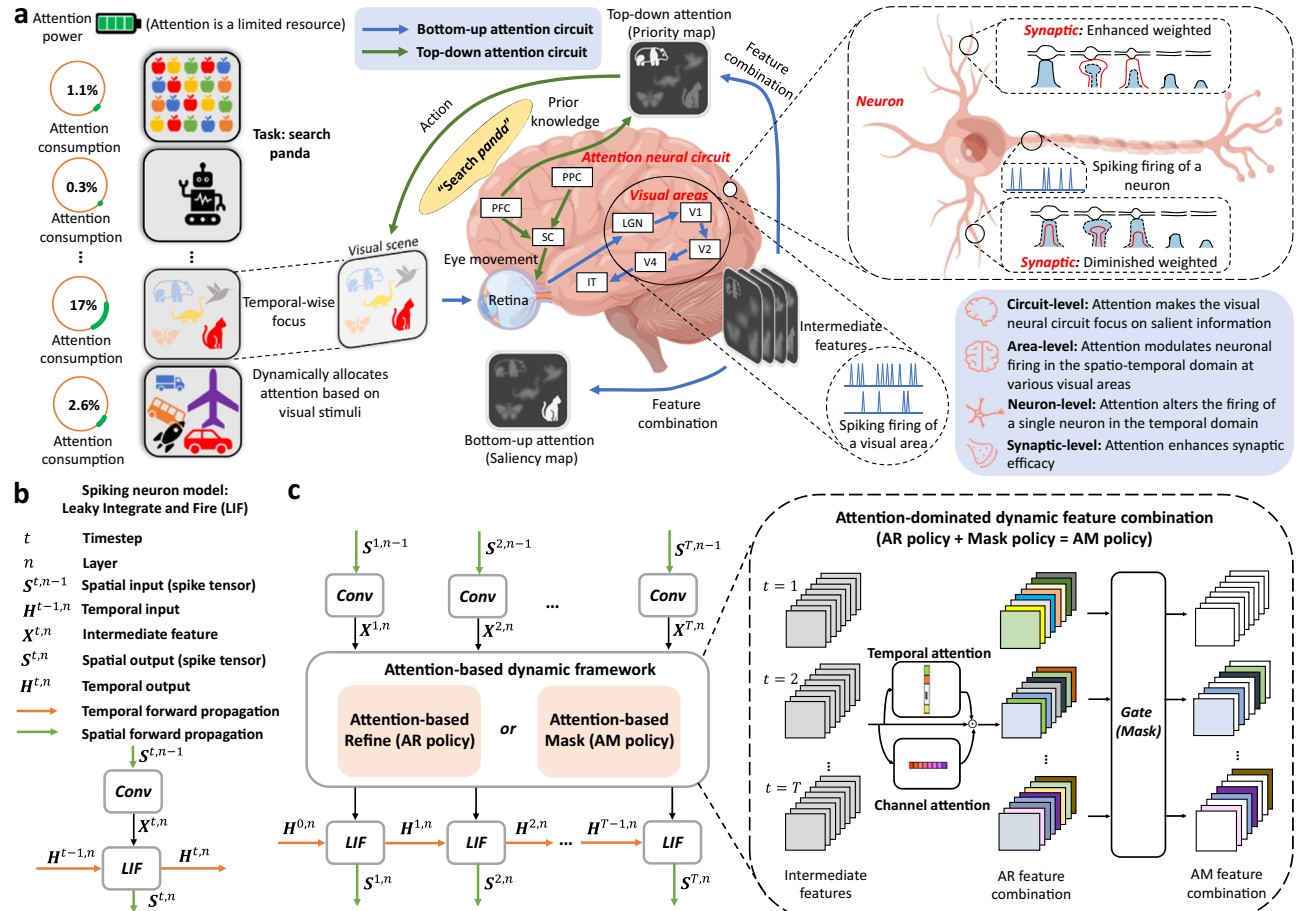

**Fig. 3 | Brain-inspired dynamic framework for neuromorphic computing.**
**a** Attention-based dynamic response in neuroscience. The brain's dynamic responses are associated with visual attention. Since attention is a limited resource, the brain only selectively processes a part of sensory input. The neural correlates of attention can be roughly divided into four structural levels[14,15]. *Attention neural circuit.* The top-down versus bottom-up dichotomy is one of the classic classifications of attention neural circuits, which encompass multiple visual areas[63]. Top-down deploys the attention to internal, behavioral goals of the brain, which can be present through the priority map. Bottom-up allocates attention according to the physical salience of a stimulus, which the salience map can illustrate. *Visual area.* The regulation of attention involves multiple brain areas, which generally results in changes in neuronal firing rate within the areas[15]. *Neuron.* Attention-related

neuronal modulations[16]. Recordings from individual cells have shown that attention is associated with the change in neuron firing, which can enhance the quality of sensory representations. *Synaptic.* Attention fine-tunes neuronal communication by selectively modifying synaptic weights, enabling enhanced detection of important information in a noisy environment[34]. **b** A typical spiking neuron model: Leaky Integrate and Fire (LIF). **c** Attention-based dynamic SNNs. The proposed dynamic framework exists as plug-and-play attention modules that optimize the membrane potential in a data-dependent manner in both temporal and channel dimensions. The dynamic framework provides two types of combinable strategies, refinement, and masking, to expand the strategy space and establish a better trade-off between accuracy and energy consumption.

dynamic SNN architecture, respectively. The proposed dynamic framework merges two classes of strategies, attention-based refinement and masking. In the refinement part, membrane potential optimization is realized by refining intermediate feature maps through the attention scores. Some not essential maps or inputs can be directly masked out in side the masking process. The dynamic framework takes the following form (see more details in "Methods" section):

$$\mathbf{U} \leftarrow \mathbf{W}_{\tau(\theta, \mathbf{U})} \odot \mathbf{U}, \tag{1}$$

where $\mathbf{U}$ is the membrane potential, $\mathbf{W}_{\tau(\theta, \mathbf{U})}$ represents the input-dependent dynamic optimization factors generated by the policy function $\tau(\theta, \mathbf{U})$, $\theta$ are parameters in the policy function, $\odot$ denotes the element-wise multiplication. $\tau(\theta, \mathbf{U})$ has a huge design space, including policy dimension and location, attention design, masking method. Optimizing the membrane potential distribution by the dynamic framework is mathematically equivalent to dynamically adjusting the weights according to the input, thus effectively mitigating the dynamic imbalance caused by the spatio-temporal invariance[25].

## Evaluation of neuromorphic system in terms of dynamic computing

We evaluate the dynamic SNN algorithm using four event-based benchmark datasets. DVS128 Gesture[30], DVS128 Gait-day[31] and DVS128 Gait-night[32] are recorded by a DVS128 camera. As the dataset's name implies, Gesture comprises hand gestures, Gait-day collects human gaits during the daytime, and Gait-night is a mated dataset for Gait-day that contains gaits at night in challenging lighting settings. Another dataset, HAR-DVS[33], is currently the largest event-based Human Activity Recognition (HAR) dataset with more than 100K samples. Some samples of these datasets are shown in Fig. 4a.

We are especially interested in what happens to the model's accuracy and spiking when we switch from vanilla to dynamic SNNs. Our framework outperforms a variety of baseline network scales (Table S5), where results on Gesture, Gait-day, and Gait-night are presented, and we exploit the same structure for all datasets, a lightweight baseline network with a three-layer Conv-based LIF-SNN[27]. We employ two attention dimensions - temporal and channel. Throughout

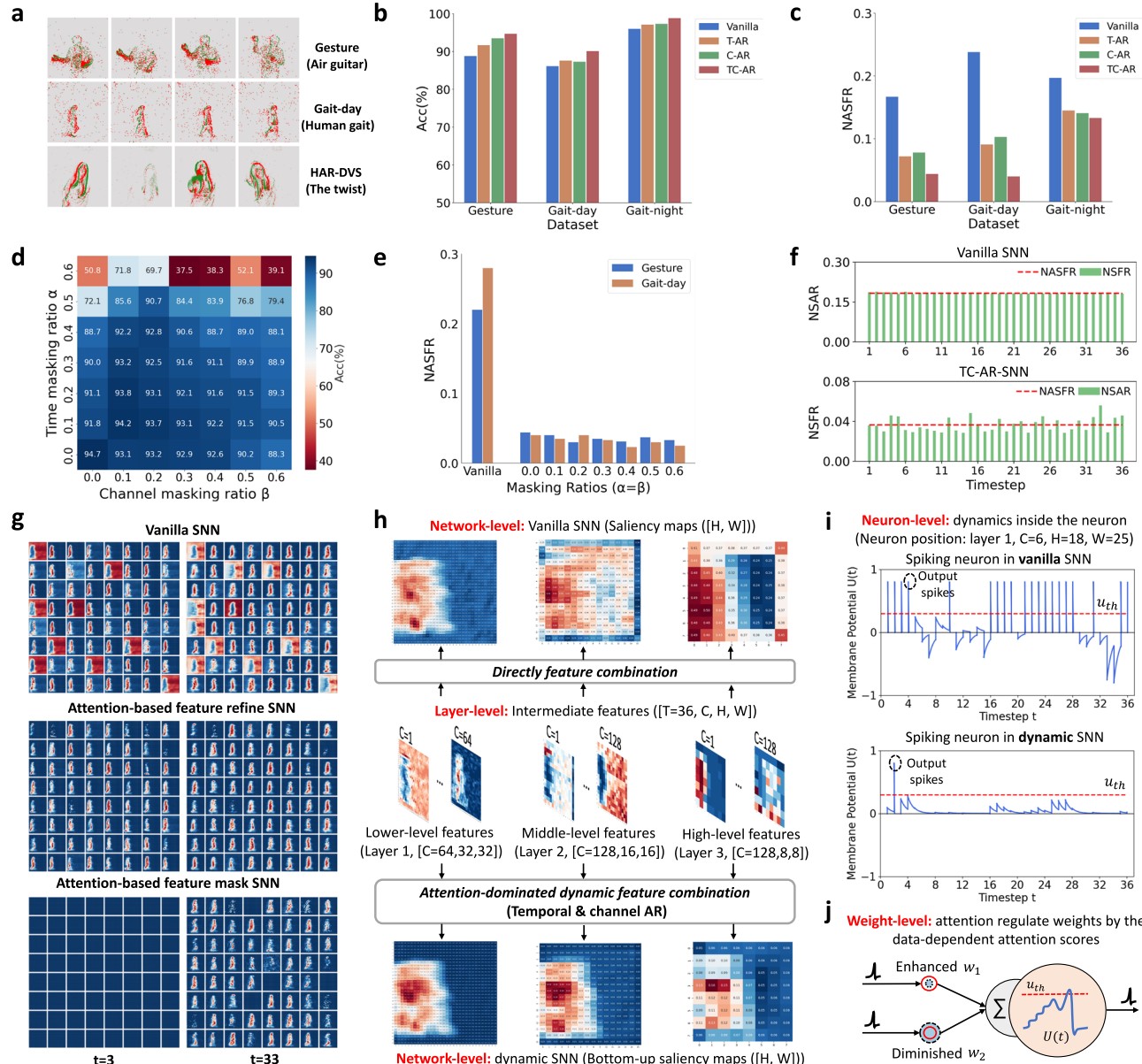

**Fig. 4 | Analysis of dynamic SNNs regarding performance and spiking activity.** **a** Examples of event-based sample. **b** Effects of Attention-based Refine (AR) policies on accuracy. Optimizing the membrane potential in both temporal and channel dimensions yields the best accuracy gain. **c** Effects of AR policies on spiking firing. Exploiting attention to optimize the membrane potential can drop spikes. **d** Effects of Attention-based Mask (AM) policies on accuracy. Increasing the masking ratios will generally result in a loss of performance. **e** Effects of AM policies on spiking firing. Adding the mask ratios does not always reduce spiking firing. **f** Spiking responses of vanilla and dynamic SNN. The proposed dynamic framework alleviates dynamic imbalance. **g** Visualization of overall spiking response on Gait. From top to bottom: spiking features in the first layer (64 channels) of vanilla, AR, and AM-SNN.

The redder the pixel, the higher NSFR (i.e., neuron spike firing rate, specific definition is given in Part "Details of algorithm evaluation"); the bluer the pixel, the closer the NSFR is to 0. Attention drives the network to focus on the target and suppress the redundant background channels, where the latter leads to a significant reduction in spikes. **h** Structural and functional correspondence between dynamic processing in neuroscience and dynamic SNNs on the network (circuit)-level and layer (area)-level. When generating saliency maps, the proposed dynamic framework recombines intermediate features according to their importance. **i** Neuronal dynamics in vanilla and dynamic SNNs. **j** Optimizing the membrane potential is mathematically equivalent to moderating the weights.

all ablation experiments, the only variable was whether the proposed module was plugged into the vanilla SNNs.

Ablation studies are conducted in Fig. 4b–e. First, we solely employ the Attention-based Refine (AR) policy on the Temporal and Channel dimensions separately or simultaneously, denoted as T-AR, C-AR, and TC-AR. We observe that refining policy can almost always improve the performance (Fig. 4b) and drop the spiking firing (Fig. 4c), whether executing on the single-dimensional or dual-dimensional. Then, the impact of the mask policy is examined. Suppose $\alpha$ and $\beta$ are two hyper-parameters denoting temporal and channel masking

proportions. On Gesture, we iterate over all possible combinations of $\alpha$ and $\beta$ range from 0 (i.e., only execute TC-AR) to 0.6, and report performance and Network Average Spiking Firing Rate (NASFR, see strict definition in the "Methods" section) in Fig. 4d, e. As mask proportions rise, we see a decline in the model's accuracy. Spiking firing, in contrast, is not sensitive to the masking proportions. Moreover, as shown in Fig. 4f, the dynamic imbalance is effectively mitigated in dynamic SNNs, and the NASFR is significantly decreased.

We explain these observations from two different angles. We begin with the entire sample set and track the variations in the spiking

firing of vanilla and dynamic SNNs in Fig. 4g. We can observe that the attention modules can adaptively inhibit the background noise (i.e., the red area, the darker the red, the higher the firing rate of spiking neurons), thereby reducing the number of spikes. Then, we examine the response of the vanilla and dynamic SNNs to the same single input sample. As depicted in Fig. 4h–j, we demonstrate that dynamic processing in neuroscience and dynamic SNNs are closely related to both structurally and functionally. Specifically, at the network (circuit)-level, we averaged the 4D ([T, C, H, W]) spiking features into a 2D map over the temporal and channel dimensions. The 2D maps represent the average spiking firing rate of SNNs and are thus considered saliency maps. Dynamic SNNs exhibit a more pronounced enhancement of the essential information region and a significant suppression of the background noise region, particularly in the final two layers. We can tell that the saliency map improves because attention suppresses some noisy intermediate channels by looking at the spiking responses of each layer (Area)-level channel. We also display the membrane potential dynamics of a single spiking neuron at the same location in both vanilla and dynamic SNNs for the identical input (neuron-level, Fig. 4i). The membrane potential of spiking neurons is optimized by attention, which influences firing.

We then evaluate dynamic SNNs on Speck. A complete neuromorphic system can be built based on Speck (Fig. 5a). The software toolchain (Figs. S8, S9, Table S3) provided by Speck makes it efficient to deploy SNN algorithms. Users only need to exploit the programmable framework Sinabs to train the model and then map it directly to Speck. We show a complete solution for employing Speck as edge computing devices for smart home application scenarios in Fig. S10. Furthermore, Speck can be easily matched with other external devices for algorithm and application testing (Fig. S11).

We test dynamic SNNs on Speck based on Gesture[30]/Gait-day[31]/Gait-night[32] datasets to facilitate comparison with other works. We modified the dynamic framework to enable the smooth deployment of dynamic SNNs. The most significant change is how to apply temporal-wise attention (Fig. 5b, c). An underlying assumption in the proposed dynamic framework in Fig. 3d is that we must first gather information at all times, forcing temporal-wise attention to only function offline. To break this assumption and fully utilize Speck's always-on, we here only employ temporal-wise attention at the input layer to judge whether event streams within a certain temporal window are needed. The temporal-wise attention scores are exploited to mask some inputs directly. For simplicity, the masking ratio in all trials is set to 0.5, which means that only half of the input is retained.

In Table 1, we show the results of deploying vanilla and dynamic SNNs on Speck and GPU regarding accuracy, power, and latency. We first focus on comparing the same SNN model running on GPU and Speck. The total power and latency required to run a single sample on Speck are much less than running on a GPU. Speck can process event streams in mW-level power with almost no latency. By contrast, running the same model on a GPU requires thousands of times more power (e.g., 30079 mW vs. 3.8 mW on Gesture) with tens of milliseconds of latency. We found that there is only a little loss of accuracy when the same model is deployed on Speck and simulated on GPU. Note, since we use public event-based datasets, the power reported in Table 1 only involves the processor part of Speck and does not include the DVS camera part of Speck.

The processor resting power is a key point that affects the total energy consumption of the hardware. The GPU consumes 30 W of power even when it is not processing any jobs, which is far more than the running power necessary to process a single sample. These experimental findings verified our earlier claim that a sufficiently low resting power is a prerequisite for dynamic computing. In contrast, neuromorphic systems based on Speck are ideal for performing dynamic computing (processor resting power is only 0.42 mW, details are given in Part "Chip performance evaluation" of "Methods" section).

Asynchronous event-driven paired with dynamic SNN input masking allows Speck to reduce energy consumption by 3× and significantly improve accuracy (Table 1). For instance, masking half of the input on Gesture drops spikes by 60.0%, reducing the total power from 9.5 mW to 3.8 mW (the lowest power sample consumes only 0.70 mW), but boosting accuracy by +9.0%.

Speck provides real-time power consumption monitoring that generates data every ms. The majority of Speck's real-time power is composed of RAM and Logic power (Fig. S6). It can be simply considered that the former is the power of data reading and neuron state update, and the latter is the power of synaptic operation. The sum of the two is the total power of the chip. We present a typical case in Fig. 5e, f, the real-time power of the same sample in vanilla and dynamic SNN. To make observation easier, we average the blurred lines on the background every 20 ms to obtain a clear solid line. On vanilla SNN, we can clearly observe the dynamic imbalance in that the real-time power changes little at all moments. Moreover, we see that the resting power of Speck is almost zero, which is fantastic for edge computing devices that must be left on for extended periods of time. In particular, when the input is masked in the dynamic SNN, Speck only costs the resting power. The power curve in Fig. 5f is more fluctuating compared to the vanilla SNN in Fig. 5e, and the peak and total power also drop greatly.

## Discussion

Although dynamic computing possesses bio-plausibility and fascinating properties such as accuracy, adaptiveness, and computational efficiency, these advantages currently exist only in theory. To truly demonstrate the power of sophisticated dynamic computing with machine intelligence, top-level design of the entire AI system is indispensable. In this work, we have shown an application-oriented algorithm-software-hardware co-designed neuromorphic system that naturally and subtly embodies the unique advantages of dynamic computing regarding energy consumption, output latency, and task accuracy. We have presented a sensing-computing asynchronous neuromorphic SoC like an eye-brain integrated system to realize attention-based dynamic computing.

At the algorithmic level, we have revealed that brain-inspired spiking communication makes SNNs inherently capable of dynamic computing. Still dynamic imbalance caused by another fundamental assumption of SNNs, spatio-temporal invariance, undermines this gift. Inspired by the attention-based dynamic response mechanism in the human brain, we proposed dynamic SNNs, which combine the attention-based dynamic framework with vanilla SNNs to improve the ability to focus on important information so that varied inputs consume energy with large variance. Experimental results show that dynamic SNNs can simultaneously achieve the two main considerations for realizing machine intelligence - effectiveness and efficiency[28]. We were pleasantly surprised to find that dynamic SNNs correspond well to the attention mechanisms in the human brain both structurally and functionally[15]. Attention stereoscopically regulates the firing of spikes in the brain's neural circuits, brain regions, and neurons, and the dynamic framework assists the vanilla SNNs in comprehensively optimizing the spike firing of networks, layers, and neurons. Consequently, dynamic SNNs can kill two birds with one stone by focusing on important information while suppressing noise spikes, significantly reducing network energy consumption while improving performance.

At the hardware level, we have demonstrated Speck, which bringing to reality the theoretical advantages of dynamic SNNs at the algorithmic level. The most intriguing feature of Speck is the low resting power (no-input consumes no running energy) brought about by the fully asynchronous design, i.e., always-on, making it particularly competitive in resource-constrained edge computing scenarios. This is also the basic hardware requirement for dynamic computing. We have demonstrated that the energy gain from the sophisticated dynamic

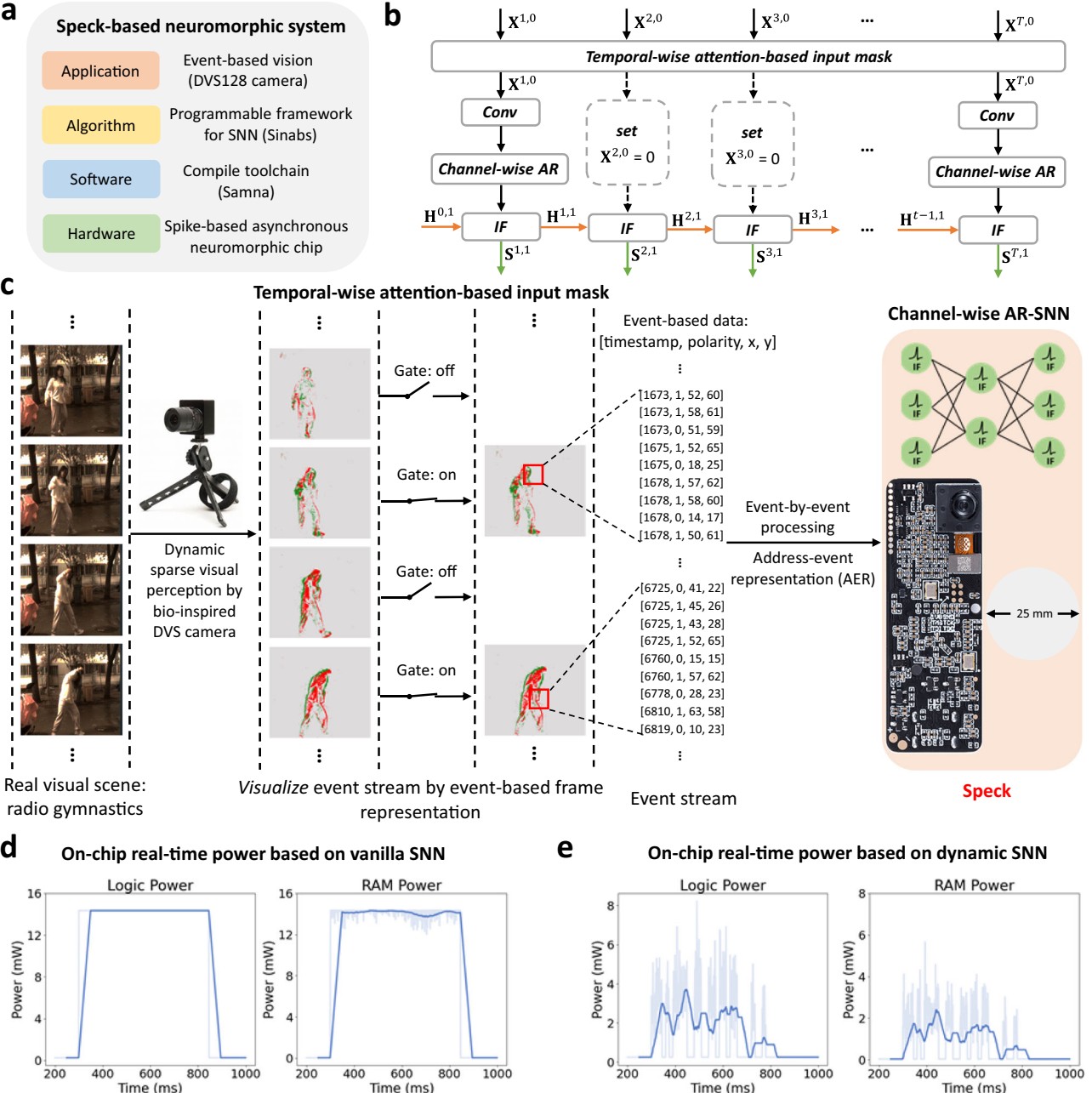

**Fig. 5 | Placement of the dynamic SNNs on Speck. a** Speck-based neuromorphic system. **b** Dynamic SNN architecture deployed on Speck. We made some algorithmic adjustments based on the proposed dynamic framework, to adapt to the hardware. We only employ temporal-wise attention on the event streams to wean out which inputs can be masked. On the other hand, the spiking neuron model on Speck is Integrate and Fire (IF), i.e., LIF neuron without leaky operation. $X^{t,n}$, $H^{t,n}$, and $S^{t,n}$ (specific definitions are given in Part "Spiking neuron models" section) represent the spatial input, temporal input, and spike output of the spiking neuron, respectively. **c** Overall of Speck-based neuromorphic system with dynamic SNNs. The DVS camera only perceives and encodes the brightness change information in

the visual scene (the red/green dots in the graph represent brightness increase/ decrease respectively.), significantly reducing spatial redundancy compared with the traditional camera. However, the high temporal resolution of the DVS causes information redundancy in the temporal dimension. We adaptively mask some inputs using temporal attention. Since Speck is event-driven, less input means lower energy consumption. Moreover, the width of Speck kit shows is equivalent to the diameter of a coin, about 25 mm, which is convenient for edge computing scenarios. **d, e** On-chip real-time power based on vanilla and dynamic SNNs, respectively.

algorithm design is completely negligible once the resting power is too high. Moreover, present-day neuromorphic computing frequently separates the design of applications, algorithms, and chips. The needs of hardware and applications are rarely considered when designing neuromorphic algorithms and vice versa. By contrast, Speck incorporates a fully asynchronous spike-based neuromorphic chip with a DVS camera, creating the perfect blend of hardware and applications well suited for dynamic computing. Calculations in Speck are only

triggered when DVS generates an event. Comprehensively, based on our top-level design of dynamic algorithms, chip architecture, and real-world application requirements, we have demonstrated mW-level power and ms-level latency solution in typical dynamic visual scenarios. This tapping into the potential of neuromorphic computing will undoubtedly advance the field.

In target edge computing environments, the overhead energy is strictly constrained, especially for a small system working in

**Table 1 | Benchmarking results**

| GPU(3090) | | | | | | | | | | |
|---|---|---|---|---|---|---|---|---|---|---|
| | **Vanilla SNN** | | | | | **Dynamic SNN (This work)** | | | | |
| **Task** | **Acc(%)** | **Rest power (mW)** | **Total power (mW)** | **Latency (ms)** | **Spike counts (×10$^6$)** | **Acc(%)** | **Rest power (mW)** | **Total power (mW)** | **Latency (ms)** | **Spike counts (×10$^6$)** |
| Gesture | 82.3 | 30000 | 30078 | 24.7 | 1.2 | 92.0 (+9.7) | 30000 | 30079 | 28.1 | 0.5 (−60.5%) |
| Gait-day | 87.2 | 30000 | 30047 | 24.5 | 2.6 | 90.2 (+3.1) | 30000 | 30048 | 27.5 | 1.0 (−60.8%) |
| Gait-night | 85.5 | 30000 | 30049 | 21.5 | 3.5 | 91.0 (+5.6) | 30000 | 30049 | 23.5 | 1.3 (−62.8%) |
| Speck | | | | | | | | | | |
| Gesture | 81.0 | 0.42 | 9.5 | <0.1 | 1.0 | 90.0 (+9.0) | 0.42 | 3.8 (−60.0%) | <0.1 | 0.4 (−60.0%) |
| Gait-day | 86.0 | 0.42 | 16.1 | <0.1 | 2.9 | 90.0 (+4.0) | 0.42 | 7.3 (−54.7%) | <0.1 | 1.2 (−58.6%) |
| Gait-night | 86.0 | 0.42 | 46.8 | <0.1 | 3.3 | 91.0 (+5.0) | 0.42 | 12.3 (−73.7%) | <0.1 | 1.5 (−54.5%) |

On the Gesture[30]/Gait-day[31]/Gait-night[32], we use exactly the same experimental settings, including input time window, network structure, training method, hyper-parameter settings, etc. We design a tiny network structure for these datasets, i.e., Input-32C3S1-32C3S2-32C3S1-AP4-32FC-Output. Note, AP4-average pooling with 4 × 4 pooling kernel size, $n$C3S$m$-Conv layer with $n$ output feature maps, 3 × 3 weight kernel size, and $m$ stride size, $k$FC-Linear layer with $k$ neurons. We first train the model on GPU (Nvidia RTX 3090) and then deploy the trained model to Speck (only five SNN cores are utilized). In the inference stage, we count the accuracy, energy consumption, and latency on both GPU and Speck. For the GPU, we set the batch size to 1, ran 1000 samples (all samples have an input time window of 540 ms), and counted their power and latency. In the whole process, we remove the consumption of the data loading process. For the Speck, we randomly sample 100 samples on each dataset as input to Speck chip to evaluate power and latency, where the total power is composed of Logic and RAM power, and the latency of a single sample is defined as the difference between the timestamp of the output result and the last input event. Thanks to the Sinabs framework and the newly proposed spiking neuron model (please see "Methods" section), the model trained on GPU has little accuracy loss after being deployed on Speck. Note, since we use public event-based datasets, the power reported in this Table only involves the processor part of Speck and does not include the DVS camera part of Speck (please see Eq. (2)).

self-powered mode for a long time. Speck is a neuromorphic chip with sensing-computing-integrated functionality, which consumes quite low-power consumption via asynchronous digital design. Such high energy efficiency and low production cost are difficult to promise modeling flexibility and computing precision. Fortunately, it is acceptable in our target scenarios where energy efficiency matters more than the task difficulty and behavior accuracy. We believe Speck can cover a broad range of neuromorphic-vision-specific edge computing tasks distinct from cloud computing while improving modeling flexibility and computing precision under the energy constraint remains an interesting and valuable direction. For example, enriching the supported network types and introducing mixed-precision computing might be possible solutions in future work.

At the software level, Speck provides a complete software toolchain to enable neuromorphic computing to be effectively and efficiently deployed in various applications based on dynamic vision. Specifically, the complete software toolchain provided by Speck, including data management, model simulation, host management, etc., can promote the rapid deployment of neuromorphic computing. We look forward to these engineering efforts to promote the advantages of neuromorphic computing in more applications.

Finally, incorporating an attention mechanism to SNNs in neuromorphic hardware can be seen as a first step towards porting more sophisticated high-level neural mechanisms in the human brain[15,34] into such hardware. As well known, neuromorphic hardware is non-von-Neumann architecture hardware whose structure and function are inspired by brains. Some unique fundamental operational characteristics, including highly parallel operation, collocated processing and memory, inherent scalability, and event-driven computation, stem from their choice to incorporate neurons and synapses to serve as the primary computational units. Although the vast majority of neuromorphic computing works have been based on the model design and hardware implementation of spiking neurons, it is unclear whether they are the only aspects of the biological brain important for performing computations. The practice in this work confirms that the attention mechanism is also very important for computing. Neuromorphic computing should consider the response of neuron granularity and perform overall control from a higher abstraction level, like the human brain. Even more exciting, these high-level abstractions of brain mechanisms may be functionally and structurally well-suited for implementation in brain-inspired neuromorphic computing. In particular, neuromorphic computing may contribute in answering one of

computational neuroscience and machine learning's important open questions: how can diverse high-level neural mechanisms generated during the evolution of the brain be imitated and incorporated into computers to enable machine intelligence to function similarly to the brain?

## Methods
### Design philosophy
Whether leveraged as small stand-alone applications or employed as edge nodes to build larger systems, edge devices call for edge hardware with unprecedented low latency and low power. In theory, neuromorphic intelligence is well suited for edge computing scenarios as it can perceive and process information sparsely. In this work, we propose a neuromorphic system to mine and cash in these unique gifts of neuromorphic intelligence through top-level co-design of hardware, algorithms, software, and applications. Our top-level design derives from two well-known common principles in computational neuroscience. First, the human brain integrates visual information from the eyes, which perceive scenes sparsely. At the same time, only a fraction of neurons in the brain respond to visual stimuli. Secondly, even if the eye perceives lots of information, some global advanced information processing mechanisms in the human brain, such as attention, will allow the brain to ignore some information to alleviate the processing burden directly.

**Eye-brain integrated hardware design.** Speck is a sensing-computing neuromorphic SoC with the spike-based sparse computing paradigm. DVS simulates biological visual pathways, asynchronously and sparsely generates events (spikes with address information) when scene brightness changes, which can drop redundant data at the source. Neuromorphic chips only activate a portion of spiking neurons to perform computations when an input event occurs (i.e., event-driven). As low-level abstractions of the human eye and brain, the sparse sensing of a single pixel in DVS and the sparse computing of a single spiking neuron in neuromorphic chip are the basic building blocks for the structure and function realization of Speck. After the physical combination of DVS and neuromorphic chip, three key designs inject soul into Speck's eye-brain integrated processing are: (a) DVS and chip interface system design, the basis of high-speed and efficient data transmission. (b) SNN convolution core design, the basis of low-latency and low-power machine intelligence. (c) Full asynchronous logic design, the basis of Speck's high-speed processing and low resting power.

**High-level brain mechanism mapping.** The dynamic response mechanism is a high-level abstraction of human brain functions, which requires global regulation of the spike firing of neurons in each brain region based on the stimulus. The human brain allocates attention according to the importance of the input. Since sensing and computing in Speck are asynchronously spike-based event-driven, at least one simple dynamic response can be realized: masking the stream of unimportant events for a period of time so that the neuromorphic chip does not perform any computations during this temporal window.

Comprehensively, the low-level abstraction of the human eye and brain is the basis for Speck's high-speed, low-latency, and low-power computing. The high-level attention abstraction can be downward compatible with the low-level abstraction, improving computational efficiency. This organic integration of multi-level brain mechanisms benefits from our firm grasp of the philosophy of brain-like spike-based sparse sensing and computing in the design of Speck.

## Chip design

We define Speck as an efficient medium-scale neuromorphic sensing-computing edge hardware. It has four main components (Fig. 2g): DVS with a spatial resolution of $128 \times 128$, sensor pre-processing core, 9-SNN cores enabling combined convolution and pooling or fully connected SNN layers, and readout core. The connection of each component mentioned is done through a universal event router. Combining such low latency, high dynamic range, and sparse sensor with an event-driven spiking Convolutional Neural Network (sCNN) processor, that excels in real-time low latency processing on a single SoC is a natural technological step. To complement the architectural advantages of always-on sparse sensing and computation, the SoC is built in a fully asynchronous fashion. The asynchronous data flow architecture provides low latency and high throughput processing when requested by sensory input while inherently shifting to a low power/idle state when the sensory input is absent. Thoroughly investigating and verifying many edge computing vision application scenarios, Speck was designed to comprise 328 K neuron units spread over nine high-speed, low latency, and low-power SNN cores.

**Sensing-computing coupling.** Current technologies exploit USB connections or other interface technologies to connect to the vision sensor and neuromorphic chips/processors. Moving data over chip boundaries and long distances impacts latency and increases power dissipation for robust signal transmission significantly. While advanced CIS-CMOS processes can couple dedicated high-quality vision sensors (vision optimized fabrication process) and neural network compute chips (logic optimized fabrication process) in a single package[35], by combining both sensor and processing on a single die into a smart sensor, we lower unit production costs significantly while saving energy on high-speed and low-latency data communication, as the raw sensory data never has to leave the chip.

**The sensor.** The sensor of Speck consists of $128 \times 128$ individually operating event-based vision pixels, also called dynamic vision pixels[20]. In contrast to the frame-based cameras, these pixels encode the incident light intensity temporally on a logarithmic intensity scale, also known as Temporal Coding (TC) encoding. In other words, the sensor transmits only novel information in the field of view, sparsifying the data stream significantly and seizing transmission on no visual changes. Each pixel is attached to a single handshake buffer to enable the pixel to work fully independently from the arbitration readout system. The arbitration is built out of one arbiter tree for column arbitration and one for rows[36,37]. The event address is encoded from the acknowledge signals of the arbitration trees and handed off as an Address Event Representation (AER) word to the event pre-processing block. The events are encoded as 1-bit polarity (ON-increasing illumination / OFF-decreasing illumination), 7-bit $x$-address, and 7-bit $y$-

address (total of 15-bit data per pixel event). A complete arbitration process with ID encoding takes approximately $2.5 - 7.5\,ns$ for a single readout. The arbitration endpoint with buffer in the pixel itself is optimized to limit the transistor count, resulting in a fill factor of 45% front illumination for each pixel. The refractory of each pixel is around 500 µs. Under the worst-case condition where the activity rate of the image is around 100%, the data needs to be transmitted during refractory is around 250 Kb ($128 \times 128 \times 15$ bit), i.e. 500 Mb/s data rate. For a typical condition, 10% to 20% of the pixel area is estimated to be active, which causes a 50 to 100 Mb/s data rate. Opposed to power-hungry high-speed low latency off-chip communication, transmitting this sparse data stream on-chip yields a significant reduction in power consumption, proportional to the data rate.

**DVS pre-processing core.** To conform the raw AER event stream from the sensor to the requirements of the sCNN, a pre-processing stage is required. The image may be flipped, rotated or cropped if only a Region of Interest (ROI) of the image is required. A lower image resolution might be required, or the polarity can be ignored. To accomplish this, the sensor event pre-processing pipeline consists of multiple stages enabling the following adjustments in the sensor event stream: polarity selection as ON only, OFF only, or both, region of interest adjustment, image mirroring in $x$, $y$ or both axes, pooling in $x$ or $y$ coordinates separately or together, shifting the origin of the image to another location by adding an offset, etc. In addition, there is also an option to filter out the noisy events coming out from the DVS by using full-digital low-pass or high-pass filters. The output of the pre-processing core has a maximum of two channels, with each channel indicating whether the event belongs to the ON or OFF category. After the pre-processing of the sensory events, the data output is transmitted to the Network on Chip (NoC).

**Network on chip.** The NoC router follows a star topology. The routing system operates non-blocking for any feed-forward network model and routes events via AER connections. The mapping system allows data to be sent from one convolution core to up to 2 other cores and for one core to receive events from multiple sources without addressing superposition with up to 1024 incoming feature channels. On every incoming channel, the routing header of every AER packet is read, and the payload is directed to the destination. This is done by establishing parallel physical routing channels that do not intersect for any network topology that does not contain recurrence. This prevents skew due to other connections and deadlocks by loops inside the pipeline structure. The routing header information is stripped from the word during transport, and the payload is delivered to its intended destination.

**SNN core.** In conventional CNN, a frame-based convolution is done. Thus, the camera's full frame must be available before starting the convolution operation. In contrast to CNN, event-driven sCNN does not operate on a full frame basis: for every arriving pixel event, the convolution is computed for only that pixel position. For a given input pixel, all output neurons that are associated with its convolution are traversed, as opposed to a kernel that is swept pixel-by-pixel over a complete image. An incoming event includes the $x$ and $y$ coordinates of the active pixel as well as the input channel $c$ it belongs to, as depicted in Fig. 2i and Fig. S4. The event-driven convolution implemented step-by-step in Speck with the following components (Fig. S5):

a. Zero padding: the event is padded to retain the layer size if needed. The image field, i.e. the address of the events, is expanded by adding pixels to the borders to retain the image size after the convolution if needed.

b. Kernel anchor and address sweep: In the kernel mapper, the event is first mapped to an anchor point in the output neuron and the kernel space. The behavior is seen in Fig. S4. Using this anchor the

kernel, represented by an address, is linked to an address point in the output space. The referenced kernel is swept over the incoming pixel coordinate. The kernel address and the neuron address are swept inversely to each other. For every channel in the output neuron space, the kernel anchor address is incremented so that a new kernel for the new output channel is used. The sweep over the kernel is repeated. If a stride is configured in either the horizontal or vertical direction, the horizontal and vertical sweeps are adjusted to jump over kernel positions accordingly.

c. Address space compression: To effectively use the limited memory space, the verbose kernel address and the neuron address are compressed to avoid unused memory locations. Depending on the configuration, the address space gets packed so that there are no avoidable gaps inside the address that are not used by the configuration.

d. Synaptic kernel memories: The kernel addresses are then distributed on the parallel kernel memory blocks according to the compressed addresses, and the specific signed 8-bit kernel weight is read. The weight and the compressed neuron address are then directed to the parallel neuron compute-in-memory-controller blocks according to the address location. Kernel positions with 0 weight are skipped during reading and are not forwarded to the neuron.

e. Neuron compute units: The compute-in-memory-controller block model an Integrate and Fire (IF) neuron with a linear leak for every signed 16-bit memory word. Besides classic read and write, the memory controller has a read-add-check_spike-write operation. Whenever the accumulated value reaches a configured threshold, an event is sent out as shown in Fig. 2h and the neuron state variable has a threshold subtraction or reset written back.

f. Bias and leak address sweep and memory: The leak (or bias) is modeled via an additional memory controller. The Leak/bias controller has a neuron individual signed 16-bit weight stored for every output channel map. An update event with this bias is sent to all its active neurons on a time reference tick. The reference tick is supplied from off-chip and is fully user-configurable.

g. Pooling: The output events are finally merged into a pooling stage. The pooling stage operates on the sum pooling principle, i.e., it merges the events from 1,2 or 4 neurons in both x and y coordinates individually.

h. Channel shift and routing: Before entering the routing NoC, the channels are shifted, and a prefix with routing information is added. One event is sent per destination for up to 2.

Speck has 9-SNN cores (layers) with different computational capabilities and memory sizing. For example, SNN core0, SNN core1, and SNN core2 have larger neuron memory sizes because the first layer, which connects to the dynamic vision sensor part, requires more neuron states with fewer input channels. As the network gets deeper, the synaptic operation or kernel memory requirement increases. The intermediate cores usually require a larger kernel memory size for generating more output channels using different kernel filters.

A key point for our presented architecture is synaptic memory utilization. Especially for CNN-based architectures, the on-the-fly computation of synaptic connections allows for minimizing memory requirements. This, in turn, saves area and energy - in the case of SRAM, both running and static. Our dedicated sCNN approach allows for many more synaptic connections by using the kernel weights stored in memory and computing all the synapses that share weights compared to standard SNN hardware implementations[8,10,38] with minimal additional compute required. As shown in Table S1, on-the-fly synaptic kernel mapping allows the deployment of bigger network models to larger feature-size CMOS, thus significantly more cost-effective fabrication technologies while matching state-of-the-art

performance. Besides exploiting SNN cores as convolutional layers, any SNN core that can be utilized as a fully connected layer with synaptic connections up to 64k, 32k, or 16k. In general, this is preferred at the last stages of the SNN.

**Readout core.** The optimal readout engine of the Speck is essential to receive the classification output directly from the chip. The last core output of the SNN can be connected to the readout engine. Unlike the neuromorphic chip's other cores (layers), only one SNN core can be connected to the readout layer for a given network configuration. The readout layer can simultaneously calculate 15 different classifications connected to 15 different output channels of the last SNN core. Each channel has a parallel processing engine that calculates the average of spike counts over 1, 16, or 32 slow clock cycles, in the range of 1 kHz to 50 kHz operation. Furthermore, an optional asynchronous internal clock is also generated by the event activity of the DVS and can be used as a timing tick of the spike count averaging function. After computing the average of each spiking channel, the average value is compared by a global threshold that is the same for all 15 readout engines. The average values that exceed this threshold are compared, and the one with the maximum value is selected as the classification winner. The index of the winner neuron or spiking channel is directly sent to the readout pins. When a network requires more than 15 classifications, the readout layer can be bypassed or not used, and the spike information of up to 1024 output feature maps can be read from the last SNN core output. To get a reasonable identification, an external processor is required to do the averaging over time and find the right class selection.

**Asynchronous logic design methodology.** DVS has $\mu$s level temporal resolution due to the asynchronous visual perception paradigm. To complement the architectural advantages of sparse sensing and computing, the neuromorphic chip in Speck is built in a fully asynchronous fashion. The asynchronous data flow architecture provides low latency, and high processing throughput when requested by sensory input, and immediately switches to a low power/idle state when the sensory input is absent. This is archived by building on the well-established Pre Charge Full Buffer low latency pipeline designs[39]. As each component is naturally gated in this design approach, no complex or slow wake-up procedures must be implemented, thereby reducing running power consumption and obtaining an always-on feature with no additional latency. Asynchronous chips make the data follow event-based timestamps rather than clock rising or falling edges. Therefore, during the idle period, there is no switching output from the DVS pixel array, and there is no information routed to the chip that leads to no running power consumed in any processing unit other than the readout layer. However, there is still a static power consumption from the DVS pixel array, and leak currents from logic and memory, which are reduced by the before-mentioned architecture resource optimizations and optional independent voltage scaling for both logic and memory.

In our asynchronous design flow, we implemented an extensive library of asynchronous data flow templates. Each function in our library is built using a 4-phase handshake and Quasi Delay Insensitive (QDI) Dual Rail (DR) data encoding, making it robust to an extended range of supply voltages, operation conditions, and temperatures[17,40]. The main functions implemented in our library are Latch/Buffer, Compose, Splits, Non-conditional Splits, Conditional Pass, and Merge functions[40]. At the last stage of the asynchronous chain, we serialize the data to be monitored to reduce the pin count. Before the serialization operation, we convert the dual rail encoded information to Bundle Data (BD) encoding. Performance is ensured by hierarchically detailed automatic floor planning that employs extensive guides and fences for the individual components and pipeline stages.

## Chip fabrication

Speck was fabricated using a 65 nm low-power 1P10M CMOS-logic process. The die layout is shown in Fig. S1. The total die size is "6.1 mm × 4.9 mm". The whole pixel array, including the dummy rows/columns and the peripheral bias circuitry, occupies an area of "2.8 mm × 3 mm". The vision sensory part consists of 128 × 128 DVS pixels using $n$-well photodiodes with a pixel pitch of 20 $\mu m$ and close to 45% of fill factor[41]. The chip's logic implementation is fully done by foundry-based standard cells. This also helps to transfer the logic implementation easily to lower or different technology nodes. For the SNN core and DVS pre-processing circuitry, a total of 7.8 Mb foundry-based Static Random-Access Memory (SRAM) cells are used. Each SNN core is designed with separate memory for its local computing unit, i.e., convolutional kernel and neuron states. Different cores do not share memory accessibility. The memory size distribution follows the standard CNN structure characteristics, which typically require higher neuron memory for the shallow layers and increasing kernel amount for deeper layers. Concerning the 128 × 128 as the largest possible input size, the biggest neuron and synaptic memory of one SNN core supporting this input size are 1.05 Mb and 0.13 Mb, respectively. Besides, each SNN core has a 16Kb bias memory for the independent channel-wise bias configuration. Each SNN core can support up to 1024 fan-in/fan-out (input/output) channels, and the kernel size can be flexibly set up to 16 × 16. The bandwidth of the core is defined as the number of Synaptic Operations (SynOp) per second that a layer can maximally process without latency, where a SynOp is defined as all the steps involved in the life-cycle of a spike arriving at a layer until it updates the neuron states and generates a spike if applied. SNN core0, the first SNN layer that typically receives the highest number of events, implements a fully symmetric parallel computation path to improve throughput up to 100 M SynOps/s. The bandwidth of other SNN cores is 30 M SynOps/s. Featured by the unique property of sCNN and LIF neuron models, the activity of the sparse sensor event stream is further reduced by every processing layer, truly exploiting the always-on only-compute-on-demand features of this architecture. The summary and performance of Speck are listed in Table S1 compared with existing neural network platforms.

## Chip performance evaluation

We here evaluate Speck's performance in terms of power and latency. Speck is an event-driven asynchronous chip whose power consumption and output latency vary with the number of SynOp.

**Power evaluation.** The total power consumption of speck contains four power rails:

$$P_{total} = \underbrace{P_{pixel,analog} + P_{pixel,digital}}_{P_{DVS}} + \underbrace{\overbrace{P_{pre} + P_{NoC} + P_{SNN} + P_{readout}}^{P_{Logic} + P_{RAM}}}_{P_{processor}}, \quad (2)$$

where $P_{DVS}$ indicates the power of DVS (see Fig. S7), $P_{processor}$ denotes the power of the Speck processor. The power consumption of the DVS pixels contains the analog part (biases and power of the analog circuits, i.e., $P_{pixel,analog}$) and the digital part (asynchronous circuits generating and routing out the events, i.e., $P_{pixel,digital}$). The power consumption of the processor also includes two parts, $P_{Logic}$ and $P_{RAM}$. Due to the split of the power tracks, we exploit the number of SynOps to measure all the power consumption of the processor, this covers all the power consumed by RAM and Logic. A SynOp includes the following steps: logic→Kernel RAM→logic→Neuron RAM Read→Neuron RAM Write→Logic. The processor's power consumption is accordingly divided into logic power (all computations for neuron dynamics, i.e., $P_{logic}$) and RAM power (read/write of kernel and neuron RAMs, i.e., $P_{RAM}$). In summary, Speck's power breakdown can be categorized into

four power tracks based on the utilization of its four modules. Power measurements are not conducted in isolation for individual functional modules but rather based on their actual usage in terms of {pixel analog, pixel digital, logic, and RAM}. Thus, $P_{Logic}$ and $P_{RAM}$ can be considered to be the sum of the power consumption of the DVS pre-processing core ($P_{pre}$), NoC ($P_{NoC}$), SNN cores ($P_{SNN}$), and readout core ($P_{readout}$).

For each power rail, the energy consumption can be divided into resting and running power, which could be measured separately in the following way:

1. Design a list of stimuli which will induce known events/SynOps rate $r_1$, $r_2$,... at the circuit under test (e.g., flickering light with fixed frequency for DVS, pseudo-random input spike stream for SNN processor, etc.)
2. Measure the average power consumption $P_1$, $P_2$,... over time for each stimulus
3. Fit a straight line to $\mathbf{P} = P_{rest} + \mathbf{r}E_{run}$. The estimated resting power is $P_{rest}$ and the running energy per spike/operation is $E_{run}$.

The power results on Speck are shown in Figs. S6, S7, and Table S2. The spiking firing rate can effectively affect power consumption with respect to event-driven processing. At a supply voltage of 1.2 V, the DVS and processor on Speck typically consume resting power consumption of 0.15 mW and 0.42 mW, respectively. In Table 1, we employ three public datasets to test the power of Speck. Since the DVS part of Speck is not exploited, the power in Table 1 is only $P_{processor}$, including the pre-processing power, the NoC power, the total computational power, and memory r/w power that is consumed by the SNN cores, and the readout power.

**Latency evaluation.** We measure the latency of the Speck by calculating the difference between an input event and an output event. Speck can be flexibly configured to include different modules in the pipeline, including the DVS pixels, pre-processing, and different SNN layers.

1. DVS response latency, defined as the time difference between the change of light intensity and the generation of the corresponding event, is measured to range from 40 μs to 3 ms, depending on the bandwidth configuration.
2. DVS pre-processing layer latency, defined as the time to adapt the raw DVS events to SNN input spikes, including pooling, ROI (Region of Interest selection), mirroring, transposition, and multicasting, is measured to be 40 ns.
3. SNN processor latency (per layer), defined as the time it takes to perform the event-driven convolution. It is measured by configuring a kernel with ones in a stride × stride square, and threshold = 1 (thus guarantees exactly one output spike is generated for every input spike). The value is related to the kernel size and the relative position of the neuron in the kernel and ranges from 120 ns to 7 μs (per layer).
4. IO delay: Speck uses a customized serial interface for event input and output. The transmission time is 32 and 26 IO interface clock cycles for spike input and output, respectively. The data are converted to/from parallel lines inside the chip for fast intra-chip transmission and processing. The conversion time, including both input and output, is 125 ns.

To evaluate the end-to-end latency of the SNN processor on Speck, we map the network in Table 1 onto the chip and record the input and output spikes. Since the chip is fully asynchronous and feed-forward, an output spike only takes place after the corresponding input spike triggers at least one spike in each layer sequentially. We measure the minimum delay between any input-output spike-pair to be 3.36 us (averaged over all samples). This includes the input spike transmission time of 1.28 us at 25 MHz IO interface clock.

## Spiking neuron models

Spiking neurons are the fundamental computation units of SNN, communicating via spikes coded in binary activations, which closely mimic the behaviors of biological neurons. The key difference between traditional artificial neurons and spiking neurons is that the latter considers the dynamics of the temporal dimension. The dynamics of a spiking neuron can be described as accumulating membrane potential over time from either the environment (via input information to the network) or from internal communications (typically via spikes from other neurons in the network); when the membrane potential reaches a certain threshold, the neuron fires spikes and updates the membrane potential. In this work, to evaluate the proposed dynamic framework and deploy SNNs on hardware for event-based tasks, we exploit three spiking neuron models, which differ in the membrane potential update and spiking firing rules.

**LIF spiking neuron.** The leaky integrate-and-fire (LIF)[42,43] model is one of the most commonly used spiking neuron models since it is a trade-off between the complex spatio-temporal dynamic characteristics of biological neurons and the simplified mathematical form. LIF neurons are suitable for large-scale SNN simulations and can be described by a differential function:

$$\tau \frac{dU(t)}{dt} = -U(t) + I(t), \tag{3}$$

where $\tau$ is the time constant, and $U(t)$ and $I(t)$ are the membrane potential of the postsynaptic neuron and the input collected from presynaptic neurons, respectively. Solving Eq. (3), a simple iterative representation of the LIF neuron[44,45] for easy inference and training can be obtained. For the convenience of describing our dynamic SNN, here we give the expression of a layer of LIF-SNN (Fig. 3b):

$$\begin{cases} \mathbf{U}^{t,n} = \mathbf{H}^{t-1,n} + \mathbf{X}^{t,n} \\ \mathbf{S}^{t,n} = \mathrm{Heaviside}(\mathbf{U}^{t,n} - u_{\mathrm{th}}) \\ \mathbf{H}^{t,n} = V_{\mathrm{reset}}\mathbf{S}^{t,n} + (\gamma\mathbf{U}^{t,n}) \odot (1 - \mathbf{S}^{t,n}), \end{cases} \tag{4}$$

where $t$ and $n$ denote the timestep and layer, $\mathbf{U}^{t,n}$ means the membrane potential which is produced by coupling the spatial feature $\mathbf{X}^{t,n}$ and the temporal input $\mathbf{H}^{t-1,n}$ (the internal state of spiking neurons from the previous timestep), $u_{\mathrm{th}}$ is the threshold to determine whether the output spiking tensor $\mathbf{S}^{t,n}$ should be given or stay as zero, Heaviside($\cdot$) is a Heaviside step function that satisfies Heaviside$(x)=1$ when $x \geq 0$, otherwise Heaviside$(x)=0$, $V_{\mathrm{reset}}$ denotes the reset potential which is set after activating the output spiking, and $\gamma = e^{-\frac{dt}{\tau}} < 1$ reflects the decay factor. Spatial feature $\mathbf{X}^{t,n}$ can be extracted from the original input $\mathbf{S}^{t,n-1}$ by fully connected (FC) or convolution (Conv) operations. When using the Conv operation,

$$\mathbf{X}^{t,n} = \mathrm{AvgPool}\left(\mathrm{BN}\left(\mathrm{Conv}\left(\mathbf{W}^n, \mathbf{S}^{t,n-1}\right)\right)\right), \tag{5}$$

where AvgPool($\cdot$), BN($\cdot$) and Conv($\cdot$) mean the average pooling, batch normalization[46], and convolutional operation respectively, $\mathbf{W}^n$ is the weight matrix, $\mathbf{S}^{t,n-1}(n \neq 1)$ is a spike tensor that only contains 0 and 1, and $\mathbf{X}^{t,n} \in \mathbb{R}^{c_n \times h_n \times w_n}$ ($c_n$ is the number of channels, $h_n$ and $w_n$ are the size of the channels).

The LIF layer integrates the spatial feature $\mathbf{X}^{t,n}$ and the temporal input $\mathbf{H}^{t-1,n}$ into membrane potential $\mathbf{U}^{t,n}$. Then the fire and leak mechanism is exploited to generate spatial spiking tensors for the next layer and the new neuron states for the next timestep. Specifically, when the entries in $\mathbf{U}^{t,n}$ are greater than the threshold $u_{\mathrm{th}}$, the spatial output of spiking sequence $\mathbf{S}^{t,n}$ will be activated, the entries in $\mathbf{U}^{t,n}$ will be reset to $V_{\mathrm{reset}}$, and the temporal output $\mathbf{H}^{t,n}$ should be decided by the $\mathbf{X}^{t,n}$ since $1 - \mathbf{S}^{t,n}$ must be 0. Otherwise, the decay of the $\mathbf{U}^{t,n}$ will be used to transmit the $\mathbf{H}^{t,n}$, since the $\mathbf{S}^{t,n}$ is 0, which means there is no

activated spiking output. After the Conv operation, all tensors have the same dimensions, i.e., $\mathbf{X}^{t,n}, \mathbf{H}^{t-1,n}, \mathbf{U}^{t,n}, \mathbf{S}^{t,n}, \mathbf{H}^{t,n} \in \mathbb{R}^{c_n \times h_n \times w_n}$.

**IF spiking neuron.** Another commonly exploited spiking neuron is integrate-and-fire (IF), a supported neuron model on Speck. IF does not leak membrane potential after spiking firing. That is, the decay factor $\gamma = 1$ in equation (4).

**M-IF spiking neuron.** When using SNNs to process event-based tasks, there is a problem of loss of accuracy after model synchronous training and asynchronous deployment (detailed later). To alleviate this problem, we designed a Multi-spike IF (M-IF) neuron model and integrated it into the programmable framework Sinabs. M-IF can be described as:

$$\begin{cases} \mathbf{U}^{t,n} = \mathbf{H}^{t-1,n} + \mathbf{X}^{t,n} \\ \mathbf{S}^{t,n} = \mathrm{M\text{-}Heaviside}(\mathbf{U}^{t,n}, u_{\mathrm{th}}) \\ \mathbf{H}^{t,n} = V_{\mathrm{reset}}\mathrm{Heaviside}(\mathbf{U}^{t,n} - u_{\mathrm{th}}) + \mathrm{MOD}(\mathbf{U}^{t,n}, u_{\mathrm{th}}) \odot (1 - \mathrm{Heaviside}(\mathbf{U}^{t,n} - u_{\mathrm{th}})), \end{cases} \tag{6}$$

where M-Heaviside($\mathbf{U}^{t,n}$, $u_{\mathrm{th}}$) is a Multi-spike Heaviside function that satisfies M-Heaviside($\mathbf{U}^{t,n}$, $u_{\mathrm{th}}) = [\mathbf{U}^{t,n}/u_{\mathrm{th}}]$ when $\mathbf{U}^{t,n} - u_{\mathrm{th}} \geq 0$, M-Heaviside($\mathbf{U}^{t,n}$, $u_{\mathrm{th}}) = 0$ when $\mathbf{U}^{t,n} - u_{\mathrm{th}} < 0$, MOD($\cdot$) is the remainder function, and [$\cdot$] is the floor function. Comparing Eq. (4) and (6), it can be seen that the main difference between the two is that $\gamma = 1$ in M-IF and multiple spikes can be fired at a timestep when the membrane potential is greater than the threshold.

To facilitate the understanding of the difference in the dynamics of these three spiking neurons, we present a simple example. Assume that at a certain timestep, the membrane potential of the spiking neuron is one, and the threshold $u_{\mathrm{th}} = 0.3$. Then, in LIF and IF, a spike is fired, and the internal state (i.e., $\mathbf{H}^{t,n}$) of the neuron is $V_{\mathrm{reset}}$; and in M-IF, three spikes are fired, and $\mathbf{H}^{t,n} = V_{\mathrm{reset}}$. Speck supports LIF and IF spiking neurons, and we exploit IF for algorithm testing.

## Synchronous training and asynchronous deployment

Due to the novel sensing paradigm of event cameras, extracting information from event streams to unlock the advantages of the camera is a challenge. Event cameras output events with $\mu$s temporal resolution, which are asynchronous and spatially sparse. One of the most common ways to meet this challenge is to convert the asynchronous event stream into other synchronous representations and then exploit corresponding algorithms for processing[20]. Specifically, when using the SNN algorithm, the event stream is aggregated into an event-based frame sequence and then sent to the network for training. The chip will perform asynchronous event-by-event processing on the input event stream by deploying the trained model to an asynchronous neuromorphic chip, thereby obtaining the minimum output latency. This training and deployment process is called "synchronous training and asynchronous deployment" (Fig. S12).

Here, we show how to convert an event stream into a sequence of frames, namely, frame-based representation. An event-based camera outputs event steam that comprises four dimensions (address event representation format): the timestamp, the polarity of the event, and two spatial coordinates (Fig. 5c). The polarity indicates an increase (ON) or decrease (OFF) of brightness, where ON/OFF can be represented via +1/-1 values. All events with the same timestamp $t'$ can form a set

$$E_{t'} = \left\{ e_i | e_i = [t', p_i, x_i, y_i] \right\}, \tag{7}$$

where $(x_i, y_i)$ indicates the coordinates and $p_i$ represents the polarity of the $i$-th event. The event set can be reassembled into an event (spike) pattern tensor $\mathbf{X}_{t'} \in \mathbb{R}^{2 \times h_0 \times w_0}$, where $h_0 = w_0 = 128$ (spatial resolution of the DVS128) and 2 is the channel number (ON/OFF). All elements in

the tensor are either 0 or 1 (the aforementioned +1/-1 is just for convenience to indicate which channel the event is on).

Assume the temporal window of two adjacent timestamps is $dt'$ ($\mu$s in DVS128), we select $\eta$ consecutive spike pattern tensors to aggregate into one frame with a new temporal window $dt = dt' \times \eta$. Formally, the event-based frame of input layer at $t$ time $\mathbf{S}^{t,0} \in \mathbb{R}^{2 \times h_0 \times w_0}$ based on $dt$ can be got by

$$\mathbf{S}^{t,0} = q(\mathbf{X}_{t'}) \quad (8)$$

where $t'$ is an integer index whose value range is $[\eta \times t, \eta \times (t+1) - 1]$, $t \in \{1, 2, \cdots, T\}$ is timestep, $q(\cdot)$ is aggregation function that could be selected[27,47] such as non-polarity aggregation, accumulate aggregation, AND logic operation aggregation, etc. We choose the accumulation function with polarity information as $q(\cdot)$ by default.

However, there is a model accuracy error between synchronous training and asynchronous deployment because the inputs during training and actual deployment are different. The inputs are frames when training on the GPU, while neuromorphic chips are processing event streams in an asynchronous event-by-event manner that updates the system state upon the arrival of a single event (Fig. S12). An effective method is to set $dt$ as small as possible during training (to reduce the difference with $dt'$) and set the timestep $T$ to be larger (to increase the total input data). But the price is a significant boost in training time.

Another effective way is to exploit the proposed spiking neuron model in equation (6). The basic idea is that the frame-based representation will increase the value of the input, and these modifications will eventually be reflected in the membrane potential of the spiking neuron. In this situation, it is irrational to give one spike at a time while disregarding the true value of the membrane potential. We, therefore, introduced the M-IF neuron that can fire multiple spikes at once depending on the membrane potential. When deploying tasks in real scenarios, we employ M-IF to train the model, and then map it to the Speck supporting the IF model. Model accuracy will scarcely be lost when a synchronously trained model is deployed asynchronously to Speck thanks to the reasonable design of $dt$ and $T$ and the use of the M-IF neuron model.

**Attention-based dynamic framework**

Our goal is to optimize the membrane potential via an attention-based module to tune spiking responses in a data-dependent manner dynamically. SNNs' building blocks, convolutional and recurrent operations, each process a single local neighborhood at a time. According to this viewpoint, the introduction of non-local operations[48] for capturing long-range dependencies is the main way for the attention mechanism to optimize membrane potential. In our dynamic framework, two types of policy functions, refinement, and masking, are covered, which can be implemented in three steps.

**Step 1: Capture global information.** (Figure S13a). Average-pooling and max-pooling are two commonly used global information acquisition methods[49,50]. The former can learn the degree information of the object, while the latter can learn the discriminative features of the object. Global information of various dimensions can be gathered by applying pooling processes in different dimensions. In a certain layer of SNN, the intermediate feature maps at all timesteps can be expressed as $\mathbf{X}^n = [\mathbf{X}^{1,n}, \cdots, \mathbf{X}^{t,n}, \cdots, \mathbf{X}^{T,n}] \in \mathbb{R}^{T \times c_n \times h_n \times w_n}$. The vectors $\mathbf{F}_{avg}, \mathbf{F}_{max} \in \mathbb{R}^{T \times 1 \times 1 \times 1}$, for instance, can be obtained by exploiting pooling compression in the temporal dimension, which represents the global information of the time dimension. Similarly, we can compress $\mathbf{X}^{t,n} \in \mathbb{R}^{c_n \times h_n \times w_n}$ to $\mathbf{F}_{avg}, \mathbf{F}_{max} \in \mathbb{R}^{c_n \times 1 \times 1}$ in the channel dimension. In addition, it is also conceivable to apply $\mathbf{F}_{avg}$ and $\mathbf{F}_{max}$ simultaneously by developing techniques to capture global information more effectively[51].

**Step 2: Model long-range dependencies.** (Figure S13a). Long-range dependencies can be modeled with the global information obtained in step 1. There are many specific modeling methods, and various methods can be designed according to the needs of specific application scenarios in terms of accuracy, computation efficiency, and parameter number[52]. The most classic method is to model long-range dependencies, namely, attention scores, through a learnable two-layer FC network[49,50]. Here, we only show this approach for simplicity. Formally, the typical two-layer attention function[50] is described by:

$$f(\cdot) = \sigma\big(\mathbf{W}_1(\mathrm{ReLU}(\mathbf{W}_0(\mathrm{AvgPool}(\cdot)))) + \mathbf{W}_1(\mathrm{ReLU}(\mathbf{W}_0(\mathrm{MaxPool}(\cdot))))\big), \quad (9)$$

where $\mathrm{AvgPool}(\cdot)$ and $\mathrm{MaxPool}(\cdot)$ represent the results of average-pooling and max-pooling respectively, the output of $f(\cdot)$ is a vector with the same dimensions as $\mathrm{AvgPool}(\cdot)$ and $\mathrm{MaxPool}(\cdot)$, $\sigma(\cdot)$ means the sigmoid function, $\mathbf{W}_0$, and $\mathbf{W}_1$ are the weights of linear layers in the shared FC.

In our framework, the input of $f(\cdot)$ is $\mathbf{X}^n$ if temporal-wise attention is performed. The pooling operations first infer the T-dimensional vectors, then the two-layer shared FC network achieves a new T-dimensional vector. Each element in this vector is an attention score, which measures the importance of the intermediate feature maps at different timesteps. For the modeling of channel-wise attention, we only need a new trainable two-layer FC network and set the input to $\mathbf{X}^{t,n}$. Consequently, we can get a $c_n$-dimensional channel-wise attention vector at each timestep. Note that channel attention vectors can be shared across the temporal dimension to reduce computation (Fig. 3c in this work), or solved independently at each timestep[28]. In terms of final task accuracy, there is little difference between the two designs.

**Step 3: Mask information.** (Figure S13b). A more aggressive policy masks part of the information based on attention scores. The value internal of the attention score in $f(\cdot)$ is (0, 1). The masking policy directly sets part of the attention scores to 0. In this work, we employ the reparameterization winner-take-all function that retains the largest top-$K$ values in $f(\cdot)$ and replaces other values assigned to 0:

$$g(\cdot) = g(f(\cdot)), \quad (10)$$

where the output dimensions of masking function $g(\cdot)$ and $f(\cdot)$ are exactly the same.

Finally, the feature maps optimized by the Attention-based Refine (AR) and Attention-based Mask (AM) modules can be expressed as:

$$\mathbf{X}^n \leftarrow f(\mathbf{X}^n) \otimes \mathbf{X}^n, \quad (11)$$

or

$$\mathbf{X}^n \leftarrow g(\mathbf{X}^n) \otimes \mathbf{X}^n. \quad (12)$$

During multiplication $\otimes$, the attention scores are broadcasted (copied) accordingly, for example, the temporal-wise attention vector is broadcasted along both the channel and spatial dimension, and the channel-wise attention vector is broadcasted along the temporal and spatial dimension. In practice, functions $f(\cdot)$ and $g(\cdot)$ have a huge design space[21,52] (Fig. S13c).

**Details of algorithm evaluation**

The attention-based dynamic framework is practiced as a plug-and-play module that can be integrated into pre-existing SNN architectures. Overall, we perform strict ablation experiments, where we add the dynamic modules to the baseline (vanilla) SNNs and keep all the rest of the experimental details consistent, including input-output coding and decoding, training methods, hyper-parameter setting, loss

function, and network structure, etc. For simplicity, we choose some baselines from previous work[27], and the typical experimental settings are given in Table S4.

**Datasets.** We test dynamic SNNs on two datasets of different scales. DVS128 Gesture[30], DVS128 Gait-day[31], and DVS128 Gait-night[32] are three smaller datasets, which are mainly designed for the application of some special scenarios. On these three datasets, we first perform algorithmic characteristics evaluation (Fig. 4) and then deploy the dynamic SNNs to Speck (Fig. 5, Table 1). On the other hand, to verify the performance of the proposed dynamic framework on deep SNNs, we conduct tests on HAR-DVS[33], which is currently the largest dataset for human action recognition. HAR-DVS records 300 categories and more than 100 K event streams with a DAVIS346 camera. The design of HAR-DVS considers many factors, such as movement speed, dynamic background, occlusion, multiple views, etc., making HAR-DVS a challenging event-based benchmark. The original HAR-DVS dataset is very large, over 4 TB. For ease of processing, the authors convert each event stream into frames via frame-based representation and randomly sample 8 frames as the official HAR-DVS dataset.

**Training details.** With the frame-based representation, both zero and non-zero integers are present in the values in the input frames $S^{t,0}$. Then the first layer of SNN is equivalent to the coding layer, which recodes these non-binary inputs into spike trains[53]. All other layers in the SNN communicate via spikes. After counting the number of spikes over the whole time window issued by the output neurons, a softmax function and cross-entropy are followed as loss functions. All models were trained using spatio-temporal backpropagation from scratch, which provides a surrogate gradient method to solve the non-differentiable problem of spike activity[44,45]. After choosing the baseline models, the general training setup is given in Table S4.

**Network structure.** In Table S5, we use three sizes of network structures for dynamic algorithm evaluation. In typical edge computing application scenarios, tasks are single and simple, so only lightweight networks are often deployed. On the Gesture, Gait-day, and Gait-night datasets, we exploit three-layer[27] and five-layer[54] Conv-based LIF-SNNs. Specifically, network structure (LIF-SNN[27]): Input-MP4-64C3S1-128C3S1-BN-AP2-128C3S1-BN-AP2-256FC-Output. Network structure (LIF-SNN[54]): Input-128C3S1-MP2-128C3S1-MP2-128C3S1-MP2-128C3S1-MP2-128C3S1-MP2-512FC-AP10-Output. AP2-average denotes pooling with $2 \times 2$ pooling kernel size, MP2-max pooling with $2 \times 2$ pooling kernel size, $n$C3S$m$-Conv layer with $n$ output feature maps, $3 \times 3$ weight kernel size, and $m$ stride size, $k$FC-Linear layer with $k$ neurons. We then employ Res-SNN-18[55] to verify the performance of the proposed dynamic framework on deep SNNs and large-scale HAR-DVS. Experimental results are given in Table S5. We can see that our algorithm significantly improves task accuracy on lightweight networks. For instance, our module improves the task accuracy on the Gesture and Gait-day datasets by 4.2% and 6.1%, respectively, when it is integrated into the three-layer Conv-based LIF-SNN. Besides, our algorithm also works well on deep SNNs, with an accuracy gain of 0.9% on HAR-DVS.

**Input time window.** When deploying event-based applications in real scenarios, it is usually assumed that a task result will be output at regular intervals, and we call this interval the input time window, i.e., $dt \times T$. Generally, the validity of the data increases and task performance improves with increasing input time window size. But the user's experience with the product will be impacted if the input time window is set too broadly. In this work, we test the performance of the proposed dynamic framework with multi-scale input time windows on the Gesture, Gait-day, and Gait-night datasets. One option is to align the $dt \times T$ to some fixed length (called restricted input). Another option is

to fix the value of $T$ and then adaptively set all $dt \times T$ to the total duration of the sample (called full input). As shown in Table S5, we set up ablation experiments on restricted input, $540\,ms = 15\,ms \times 36$. Moreover, in Gesture, Gait-day, and Gait-night, the duration of each sample is about 6000 ms, 4400 ms, and 5500 ms, respectively. We also test the performance of vanilla and dynamic SNNs on full input. Experimental results show that dynamic SNNs performs better than vanilla SNNs in both accuracy and network average spiking firing rate at multiple input time window scales (Table S5).

**Theoretical energy consumption evaluation.** In traditional ANNs, the times of floating-point operations (FLOPs) are used to estimate computational burden, where almost all FLOPs are MAC[56]. Following this line of thinking, in SNNs, the theoretical energy consumption in the network is also evaluated by FLOPs[28,55,57,58], which are mainly AC operations and are strongly correlated with the number of spikes. Although this evaluation view ignores the hardware implementation basis and the temporal dynamic of spiking neurons[59], it is still useful for simple analysis and evaluation of algorithm performance and guidance for algorithm design. We employ this method to qualitatively analyze the energy consumption of dynamic SNNs.

In general, the energy consumption of SNNs is related to spiking firing, and energy efficiency improves with smaller spike counts. We define three kinds of spiking firing rates to comprehensively evaluate the impact of the proposed dynamic framework on firing. Firstly, we define Network Spiking Firing Rate (NSFR) as: at timestep $t$, a spiking network's NSFR is the ratio of spikes produced over all the neurons to the total number of neurons in this timestep. Then, we define the Network Average Spiking Firing Rate (NASFR) as the average of NSFR over all timesteps $T$. NSFR is used to evaluate the change of the same network's spike distribution at different timesteps; NASFR is exploited to compare the spike distribution of different networks. Finally, we define the Layer Average Spiking Firing Rate (LASFR) to finely evaluate the energy consumption of SNNs. At timestep $t$, a layer's spiking firing rate (LSFR) is the ratio of spikes produced over all the neurons to the total number of neurons in that layer; then we define the LASFR that averages LSFR across all timesteps $T$. It should be noted that a set of NSFR, NASFR, and LASFR can be obtained for each sample. In this work, by default, we count these spiking firing rates of the samples in the entire test set and get their mean as the final data.

Specifically, the inference energy cost of vanilla SNN $E_{\text{Base}}$ is computed as

$$E_{\text{Base}} = E_{\text{MAC}} \cdot FL^1_{\text{SNNConv}} + E_{\text{AC}} \cdot \left( \sum_{n=2}^{N} FL^n_{\text{SNNConv}} + \sum_{m=1}^{M} FL^m_{\text{SNNFC}} \right), \quad (13)$$

where $N$ and $M$ are the total number of layers of Conv and FC, $E_{\text{MAC}}$ and $E_{\text{AC}}$ represent the energy cost of MAC and AC operation, $FL^n_{\text{SNNConv}} = (k_n)^2 \cdot h_n \cdot w_n \cdot c_{n-1} \cdot c_n \cdot T \cdot \Phi^{n-1}_{\text{Conv}}$ and $FL^m_{\text{SNNFC}} = i_m \cdot o_m \cdot T \cdot \Phi^{m-1}_{\text{FC}}$ are the FLOPs of $n$-th Conv and $m$-th FC layer, respectively, $k_n$ is the kernel size of the convolutional layer, $I'm$, and $o_m$ are the input and output dimensions of the FC layer, $\Phi^n_{\text{Conv}}$ and $\Phi^m_{\text{FC}}$ are the LASFR of SNN at $n$-th Conv and $m$-th FC layer. The first layer is the encoding layer, thus its basic operation is MAC. Refer to previous work[58,60], we assume the data for various operations are 32-bit floating-point implementation in 45 nm technology[61], in which $E_{\text{MAC}} = 4.6\,pJ$ and $E_{\text{AC}} = 0.9\,pJ$.

It can be seen from equation (13) that once the network structure and the simulation timestep $T$ are determined, $E_{\text{Base}}$ is only related to the spiking firing rate, i.e., $\Phi^n_{\text{Conv}}$ and $\Phi^m_{\text{FC}}$. In the case of dynamic SNNs, we optimize the membrane potential using the attention module, which in turn drops the spiking firing. So the energy increase comes from MAC operations due to the regulation of membrane potential. The energy decrease comes from the drop of AC operations caused by

sparser spiking firing. That is,

$$\Delta_E = E_{\text{Dyn}} - E_{\text{Base}} = E_{\text{MAC}} \cdot \Delta_{\text{MAC}} - E_{\text{AC}} \cdot \Delta_{\text{AC}}, \qquad (14)$$

where $E_{\text{Dyn}}$ is the energy cost of dynamic SNN, $\Delta_{\text{MAC}}$, and $\Delta_{\text{AC}}$ are the number of operations increased to compute the attention scores and reduced in the SNN, respectively. The energy consumption of dynamic SNN is lower than that of vanilla SNN as long as $\Delta_E < 0$. Further, we can define the relative energy change ratio $r_{\text{REC}}$ as:

$$r_{\text{REC}} = \frac{\Delta_E}{E_{\text{Base}}}. \qquad (15)$$

Equation (13) indicates that a sparse firing regime is the key to achieving the energy advantage of SNN. The ideal situation is that we can get the best task performance with the fewest spike, but the relationship between spike firing and performance is complex and has not been adequately explored. This work demonstrates that data-dependent attention processing can significantly improve performance while substantially reducing the number of spikes.

As shown in Eq. (14), we pay a price in the attention-based dynamic process, but overall the benefits outweighed the costs. Empirically, the firing of SNNs will be related to the dataset and network size. For example, on Gesture and Gait-day datasets, the network average spiking firing rate of the three-layer is greater than that of the five-layer Conv-based LIF-SNN (Table S5). Consequently, the reduction in the number of spikes of the three-layer SNN is also more obvious, e.g., on the Gait-day, the AM module drops the spiking firing rate by 87.2% resulting in $r_{\text{REC}} = -0.68$. By contrast, in five-layer SNN, the spiking firing rate of dynamic SNN reduced by 60.0%, while $r_{\text{REC}} = -0.18$ (there are too many MAC operations in the first encoding layer of baseline SNN that leads to higher $E_{\text{Base}}$). Thus, we can observe that $r_{\text{REC}}$ is associated with the structure of the baseline. These analyses inspire us that when designing dynamic SNN algorithms for practical applications, we should comprehensively consider factors such as task scale, network scale, attention module design, spiking firing, task performance, etc., in order to achieve the highest task cost-effective.

**Comparison with previous methods.** In Table S5, we make a comparison with prior work. On the Gesture, Gait-day, and Gait-night datasets, we can get state-of-the-art or comparable results. Since we mainly want to verify the dynamic algorithm, we did not blindly expand the network scale on these datasets. Compared with other types of networks that process event streams, such as Graph Convolutional Networks (GCN), SNNs have obvious performance advantages when the input time window is limited. For example, we achieve an accuracy of 98.2% with 540 ms on Gait-night, which is higher than 94.9% obtained by 3D-GCN using the full input[32]. We also tested the accuracy of dynamic SNNs when restricting the input time window to 120ms on the Gait-day and Gait-night datasets. The performance advantage of dynamic SNN is more significant in this case. For instance, we get an accuracy of 84.8% on Gait-day, which outperforms about 38.5% with 200 ms in 3D-GCN[32]. Due to the lack of performance benchmarks in the SNN domain for HAR-DVS. We first use Res-SNN-18[55] as a baseline and then incorporate our modules. Our dynamic SNN achieves an accuracy of 46.7% on HAR-DVS, which is close to the performance results obtained using ANN methods[33]. Although it is beyond the scope of this work, we anticipate that further effectiveness and efficiency gains will be achievable simultaneously by tailoring attention-based dynamic module usage for specific event-based tasks.

In addition, by observing the results in Table S5, we pose a valuable but complex open question: what factors will and how they affect the spiking firing rate of SNNs? Empirically, the firing of the network is

related to dataset size, network scale, input time window and timestep, spiking neuron types, etc. For instance, for the Gesture dataset, the NASFR of the lightweight three-layer LIF-SNN[27] (0.17) is significantly larger than that of the larger five-layer LIF-SNN[54] (0.07). Consequently, once the NASFR of a vanilla SNN is already low, it is relatively hard to drop spiking firing further. These observations are crucial for understanding the sparse activations of SNNs, and we look forward to their theoretical and algorithmic inspiration for subsequent research.

**Deployment of SNNs on Speck**
Speck greatly eases the difficulty of deploying SNNs on neuromorphic chips by providing a complete toolchain (Figs. S8, S9). First, we offer Tonic, a data management tool that enables users to manage existing/handcraft event data. Tonic provides efficient APIs that allow users' easy access to various available public event camera datasets. Then, to facilitate chip-compatible neural network development, we propose the Sinabs Python package. Sinabs is a deep learning library based on PyTorch for SNNs, with a focus on simplicity, fast training, and scalability. It supports various SNN training methods, such as ANN-to-SNN conversion[62], direct training method based on Back Propagation Through Time (BPTT)[44], etc. It also allows the user to freely define the synapse/spiking neuron model, and surrogate gradient function to enable advanced SNN development. Sinabs also comes with many useful plugins. For instance, with the Sinabs-Speck plugin, the user can easily quantize and transfer the model parameter and generate the compatible configuration for Speck; the EXODUS plugin optimizes the gradient flow and can accelerate the BPTT training 30 × faster.

Moreover, we provide Samna, the developer interface to the toolchain, and run-time environment for interacting with Speck. Developed towards efficiency and user friendly, a set of Python APIs is available in Samna with the core running in C++. Thus, users can utilize neuromorphic devices professionally and elegantly. Samna also features an event-based stream filter system that allows real-time, multi-branch processing of the event-based stream coming in or out from the device. With an integration of a just-in-time compiler in Samna, the flexibility of this filter system has been taken to an even higher dimension, which supports adding users' defined filter functions at run-time to meet the requirements of any different scenarios.

## Data availability
All data used in this paper are publicly available and can be accessed at https://research.ibm.com/publications/a-low-power-fully-event-based-gesture-recognition-system for DVS128 Gesture dataset, https://github.com/zhangxiann/TPAMI_Gait_Identification for DVS128 Gait-day/Gait-night datasets. The HAR-DVS dataset is available on request via https://github.com/Event-AHU/HARDVS.

## Code availability
The source code is publicly available at https://github.com/BICLab.

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

## Acknowledgements

This work was partially supported by National Science Foundation for Distinguished Young Scholars (62325603), Beijing Natural Science Foundation for Distinguished Young Scholars (JQ21015), National Natural Science Foundation of China (62236009, U22A20103, 62332002, 62027804), and Major Scientific and Technological Innovation Project of Xianyang (No.L2023-ZDKJ-JSGG-GY-018). We would like to thank Dr. Dylan Richard Muir and Dr. Sadique Sheik for their work on software toolchain, and the other members of the chip design and software toolchain design team. We also thank Huanhuan Gao (Xi'an Jiaotong University), Hengyu Zhang (Tsinghua University), Jiakui Hu (Peking University), Yuhong Chou (Xi'an Jiaotong University) for the discussion of algorithm design, and Prof. Huihui Zhou (Pengcheng Lab) and Prof. Zhengyu Ma (Pengcheng Lab) for providing the computing resources, and Baiyu Chen (Dalian University of Technology), Siyu Ding (Tsinghua University), Siyuan Xu (Shanghai Jiaotong University) for the check and revise of the manuscript.

## Author contributions

M.Y., G.L., N.Q., G.Z., B.X., and Y.T. conceived the work. M.Y. is the designer of the algorithm and carried out the simulation experiments. N.Q. is the director of the hardware and software toolchain design team. O.R., N.Q., T.D., M.D.M, C.N, S.S., and C.W. contributed to Speck manufacturing and design. Y.X., T.H., M.Y., D.W., and W.F. carried out the hardware implementation. All of the authors contributed to the discussion of the experiment result analysis, and G.L. led the discussion. M.Y. and G.L. wrote the manuscript, with additional contributions by O.R. The whole project is supervised by G.L. All authors reviewed the manuscript.

## Competing interests

The authors declare no competing interests.
