## [Peer Review File · Nature Communications]

REVIEWER COMMENTS

Reviewer #1 (Remarks to the Author):

This work implemented a spike-based neuromorphic chip with software-hardware co-designed. The author designed a neuromorphic chip called “Speck”, integrated with a dynamic vision sensor (DVS), and asynchronous neuromorphic chip achieving low resting power. Also, the author provides a complete framework for data management, simulation... for the Speck. This work provides high efficiency, low latency, and high performance with event-driven scenarios. There are several concerns and details that should be addressed and provided.

1. The author's claim of achieving a power consumption of 0.7mW and low latency of less than 0.1ms on a single sample in public datasets is indeed impressive. However, to provide a more comprehensive understanding of this achievement, the author should specify the exact conditions under which these results were obtained. This should include details about the specific dataset used, the neural network model employed, and any relevant parameters and setup configurations.
2. Furthermore, it is crucial to clarify which components or subsystems of the overall power consumption are included in this measurement. For instance, the author should delineate whether the stated power consumption encompasses the SNN core, memory buffer, dynamic vision sensor (DVS), or any other relevant elements.
3. A breakdown of power usage within the system will offer a more precise insight into the chip's energy-efficient operation and its strengths in handling diverse computational tasks. Could the author kindly provide a power breakdown of the system?
4. The author's assertion of zero running power consumption from logic gates during idle periods is a notable claim that requires further elucidation. To provide a more comprehensive understanding, the author should expound upon the techniques and mechanisms used to accomplish this feat. Specifically, a discussion of power-gating methods, the granularity of power gating, and control mechanisms is essential. Addressing the management of leakage power, including strategies, process technologies, and monitoring, is equally important. Additionally, the author should quantify and discuss the energy and time overhead associated with waking up the chip from its idle state.
5. In Table 1, it seems that the accuracy loss of Gesture on Speck using dynamic SNN is much more significant (2%) than running on GPU, compared to other cases. Could the author kindly discuss the key points leading to the phenomenon? Also, the author mentions the power and latency is running under 1000 samples. Is the accuracy also obtained from this condition or running under whole datasets?
6. In Figure 1c, what does network spiking firing rate at each timestep mean? Why does the NSFR of each layer accumulate on the same bar? Could the author kindly explain it?
7. It seems that this paper lacks of some details of how to implement the SNN cores. Could the author kindly add some details of implementation?

Reviewer #3 (Remarks to the Author):

The document details the development of "Speck", an innovative neuromorphic chip, emphasizing its low-power consumption and dynamic computing abilities. The neuromorphic chip "Speck" showcases a novel design approach, potentially advancing the field of neuromorphic computing. The paper provides detailed technical descriptions of the chip's architecture and functions. The work contributes to the ongoing research in low-power, brain-inspired computing, which is a significant and growing area of interest in the field of computer science.

I recommend addressing the following issues:

1- The description of the neuromorphic chip's design and function could be more detailed. It's unclear how certain design choices impact the chip's performance, which might lead to questions about the reproducibility and scalability of the results.

2- The paper could benefit from a more robust experimental section. Specifically, comparisons with other neuromorphic chips or standard computing systems would provide a clearer context for the chip's performance and efficiency.

3- The paper sometimes generalizes its findings without acknowledging the specific context or limitations of the study. This could lead to an overestimation of the chip's applicability in diverse computing environments. For instance, it suggests that the chip's efficiency and low-power consumption could revolutionize how computing tasks are performed, especially in energy-constrained environments.

Overall, while the paper presents an interesting and potentially significant advancement in neuromorphic computing, strengthening these areas could enhance its scientific rigor and credibility. However, the paper could benefit from a more structured discussion section that directly addresses potential limitations and broader implications of the research.

Response to the comments of Manuscript# NCOMMS-23-49454A

ID: NCOMMS-23-49454A

Title: Spike-based dynamic computing with asynchronous sensing-computing neuromorphic chip

Authors: Man Yao, Ole Richter, Guangshe Zhao, Ning Qiao, Yannan Xing, Dingheng Wang, Tianxiang Hu, Wei Fang, Tugba Demirci, Michele De Marchi, Lei Deng, Tianyi Yan, Carsten Nielsen, Sadique Sheik, Chenxi Wu, Yonghong Tian, Bo Xu, Guoqi Li

We would like to sincerely thank all the Reviewers for their careful reading and reviewing, which have helped us a lot to modify the paper accordingly. Please refer to our detailed point-to-point answers in the following response and the revisions in the revised manuscript.

Please note that reviewers’ comments are in *bold italics*. Modifications in the manuscript are highlighted in blue for your convenience.

1 Summary of Changes

We first summarize the main changes below to give a bird view of the revision for the convenience of Reviewers. We then provide point-to-point responses to Reviewer comments.

(1) Details of Speck’s power evaluation on public datasets. The test details for Speck power estimation are systematically provided. These include the introduction of three event-based public datasets that have been tested on Speck processor, instructions for training models on GPUs that can be deployed on Speck, an experimental setup for network architecture, Speck deployment instructions for the trained model, an experimental setup for power evaluation, and an experimental setup for latency evaluation. Specifically, **Speck’s resting power consists of the DVS and the processor, which are 0.15mW and 0.42mW** respectively (we provide detailed testing procedures in the supplementary materials.). In Table 1 of the main text, since we used public Gesture/Gait-day/Gait-night datasets and did not use Speck’s DVS part, the power in Table 1 only involves the power of the Speck processor. Please refer to the response to Reviewer #1’ Comment 1 for more details.

(2) Speck power consumption composition. We combined the Speck system architecture figure (**Figure 1 in this response**) to respond to the power components of Speck in detail, and provided the power breakdown of Speck through two additional Tables (**Table 3 and Table 4 in this response**). Specifically, the chip’s power breakdown can be categorized into four power tracks based on the utilization of its four modules. Power measurements are not conducted in isolation for individual functional modules but rather based on their actual usage in terms of {pixel analog, pixel digital, logic, RAM}. In the resting state, the powers of these four modules are $71.50\mu\text{W}$, $78.07\mu\text{W}$, $360\mu\text{W}$, $63\mu\text{W}$ respectively. The first two modules constitute Speck’s resting power in the DVS part, i.e., $71.50\mu\text{W} + 78.07\mu\text{W} = 0.149\text{mW} \approx 0.15\text{mW}$. The last two modules constitute Speck’s resting power consumption in the processor part, i.e., $360\mu\text{W} + 63\mu\text{W} \approx 0.42\text{mW}$. Please refer to the response to Reviewer #1’ Comments 2 and 3 for more details.

(3) Chip design details. Based on the comments of two Reviewers, we have deeply modified the chip design part to make it easier for readers to understand the origin of Speck’s low power consumption. Specifically, we revised **Part “Chip design” in Materials and Methods** very carefully to better show the design details. The revised paragraphs are as follows, **“chip design” overview**, **“Sensing-**

computing coupling”, the “Sensor”, “DVS pre-processing core”, and “Chip fabrication”. In order to specifically cover the router mechanism in Speck, we have added a new paragraph “**Network on Chip**”. We have rewritten the subsection “SNN Core” in a step-by-step breakdown of the operation of the SNN core and have added a new Figure (**Figure S5, page 30 in the main text, Figure 4 in this response**) to provide assistance in understanding the operation of the SNN core. Moreover, the “Asynchronous logic design methodology” part is enriched, to provide a more comprehensive understanding of Speck’s low resting power. Please refer to the response to Reviewer #1’ Comments 4, 7 and Reviewer #3’ Comment 1 for more details.

(4) Discussion about the limitations of this work. To explore the potential limitations and broader implications of this work, we have **added the following paragraph to the Discussion section**. “In target edge computing environments, the overhead energy is strictly constrained, especially for a small system working in self-powered mode for a long time. Speck is the first neuromorphic chip with sensing-computing-integrated functionality, which consumes quite low power consumption via asynchronous digital design. However, such high energy efficiency and low production are difficult to model with flexibility and computing precision. Fortunately, it is acceptable in our target scenarios where energy efficiency matters more than the task difficulty and behavior accuracy. We believe Speck can cover a broad range of neuromorphic-vision-specific edge computing tasks distinct from cloud computing, while improving modeling flexibility and computing precision under the energy constraint remains an interesting and valuable direction. For example, enriching the supported network types and introducing mixed-precision computing might be possible solutions in future work.” Please refer to the response to Reviewer #3’ Comment 3 for more details.

(5) Additional details need clarification. We analyzed the reasons for the performance loss of the same SNN model when running on GPU and Speck, including model deployment, training, datasets, algorithm, and testing (Reviewer #1’ Comment 5). We explained Fig. 1c in the main text as well as the definition of spiking firing rate. In order to better demonstrate the changes in spike firing in each layer, we redraw Fig. 1c (**Fig.2 in this response**) (Reviewer #1’ Comment 6). In addition, we presented Speck’s energy consumption estimation method and comparison with other neuromorphic chips (Reviewer #3’ Comment 2).

2 Response to Reviewer #1

This work implemented a spike-based neuromorphic chip with software-hardware co-designed. The author designed a neuromorphic chip called “Speck”, integrated with a dynamic vision sensor (DVS), and asynchronous neuromorphic chip achieving low resting power. Also, the author provides a complete framework for data management, simulation... for the Speck. This work provides high efficiency, low latency, and high performance with event-driven scenarios. There are several concerns and details that should be addressed and provided.

General Response to Reviewer #1. We greatly appreciate you for spending valuable time in providing insightful comments and suggestions on the manuscript. This truly helped us to improve it. We have considered your questions very carefully and have responded point-to-point. In addition, we have revised the main text accordingly to be consistent with the response letter. We hope that you will find our responses satisfactory. We will be happy to supply any technical details you require.

Comment 1: *The author’s claim of achieving a power consumption of 0.7mW and low latency of less than 0.1ms on a single sample in public datasets is indeed impressive. However, to provide a more comprehensive understanding of this achievement, the author should specify the exact conditions under which these results were obtained. This should include details about the specific dataset used, the neural network model employed, and any relevant parameters and setup configurations.*

Answer to Comment 1: Many thanks for the careful review. In this work, we aim to introduce the proposed neuromorphic system as completely as possible in one paper. Some experimental details may be scattered in different parts of the paper due to the multiple aspects of applications, algorithms, and hardware involved. This may confuse you, and we apologize for that. Here, we start from the perspective of hardware testing and systematically introduce how we perform accuracy and energy consumption testing on public datasets on Speck.

1) Datasets. As we introduced in the “**Details of algorithm evaluation**” of the “**Materials and Methods**” (page 18), We exploited the public event-based datasets DVS128 Gesture [1], DVS128 Gait-day [2], and DVS128 Gait-night [3] for Speck testing. All three datasets were recorded by the DVS128 camera, which has the temporal resolution at μs level and 128×128 spatial resolution. They differ in details. Gesture contains 11 hand gestures from 29 subjects under three illumination conditions. It records 1342 samples of 11 gestures, each with an average duration of 6 seconds. Gait-day contains various gaits from 21 volunteers (15 males and 6 females) under two kinds of viewing angles. It records 4200 samples, and each gait has an average duration of 4.4 seconds. The Gait-night dataset is recorded to investigate if the event camera can capture human gaits in low-light conditions. Gait-night contains various gaits from 20 volunteers, and each volunteer contributed 200 gait samples.

2) Training Details on GPU. We cover algorithm training details in the Part “**Details of algorithm evaluation (page 18)**”. For the sake of brevity, we omit some details and simply state that we follow the training setup from our previous work [4]. Here we explain in detail **how we train a model on the GPU that can be deployed on Speck**, which mainly includes four parts: data pre-processing, M-IF spiking neuron, data augmentation, and network structure. Among these, data pre-processing and M-IF neuron are designed to solve the problem of synchronous training and asynchronous deployment.

Hyper-parameter	Gesture/Gait-day/Gait-night
Temporal Resolution dt	15
Simulation Timestep T	36
Spatial Resolution	32×32
Spike Firing Threshold	0.3
Temporal mask ratio of dynamic SNN	0.5
Batchsize	64
Epochs	100
Weight decay	1e-4
Learning rate	5e-4
Learning algorithm	STBP
Loss function	CrossEntropyLoss
Optimizer	Adam

Table 1: Hyper-parameter setting.

2.1) *Data pre-processing.* The original event stream is asynchronous, and its temporal resolution is microseconds. It is not feasible to train the model directly with ms temporal resolution on the GPU because it would mean huge simulation time steps. A standard processing method [5] in the field is to divide the event stream within a certain temporal window into continuous event frames (details in **“Synchronous training and asynchronous deployment” subsection of the “Materials and Methods” (page 16)**). This way, we can handle the event stream like a video. This is the so-called “synchronous training”; the key hyper-parameters involved are Temporal Resolution dt and simulation Timestep T . In this work, for the sake of simplicity, for models that need to be deployed on Speck, we uniformly set $dt = 15ms$ and $T = 36$. Therefore, Speck will output the classification results every $540ms$. Additionally, event frames are downsampled to 32×32 before entering the network.

2.2) *M-IF Spiking neuron.* When training SNN on GPU, the input is $T = 36$ event-based frames. When deploying the trained SNN to Speck, the input is an event stream with a temporal window of $540ms$. This is the so-called “asynchronous deployment” (**please see Figure S11**). To reduce the accuracy error due to synchronous training and asynchronous deployment, we use M-IF when training on the GPU. M-IF and IF only differ in the way they fire (**Eq.(6) in page 16**). M-IF supports firing multiple spikes at a single timestep, which, to a certain extent, can offset the loss of temporal information caused by the aggregation of event streams into frames.

2.3) *Data augmentation.* On these DVS datasets, the duration of each sample is a few seconds, but during training, we only intercept $540ms$ of them to train the network. This is unreasonable. Therefore, we use the Random Consecutive Slice (RCS) data augmentation proposed by our previous work [4]. That is, in an event stream sample, we randomly select the starting point of the input and intercept $540ms$ to train the network.

2.4) *Network structure.* As mentioned **in the caption of Table 1 (page 7)**, we design a tiny network structure for these datasets, i.e., Input-32C3S1-32C3S2-32C3S1-AP4-32FC-Output. Note, AP4-average pooling with 4×4 pooling kernel size, $nC3Sm$ -Conv layer with n output feature maps, 3×3 weight kernel size, and m stride size, kFC -Linear layer with k neurons.

2.5) *Other training details.* We exploit standard SNN training methods, including learning algorithms (STBP), loss function, and other standard training techniques of deep learning, such as batch normalization, dropout, etc.

	Input	Output	Kernal	Stride
Conv-1	2	32	3	1
Conv-2	32	32	3	2
Conv-3	32	32	3	1
AvgPool			4	4
FC-1	512	32		
FC-2	32	11(20)		

Table 2: Network structure. Note that Gesture has 11 categories, and Gait-day/night have 20 categories.

	Power
Resting Power of Speck (excluding DVS)	0.42 mW
Resting Power of DVS on Speck	0.15 mW
Running inference power (excluding DVS) *	0.28 mW
Resting Power of Speck (including DVS)	$0.42 + 0.15 = 0.57$ mW
Total inference power of a Speck (excluding DVS)	$0.42 + 0.28 = 0.70$ mW

Table 3: Power Consumption Measurement of Speck. * Lowest single sample.

The corresponding hyper-parameters are shown in Table 1. The network structure of the three tasks is the same, see Table 2. We adopt the same baseline and hyper-parameters for vanilla and dynamic SNN, the only difference is that dynamic SNN plugs additional attention modules. The temporal-wise attention scores of dynamic SNNs are exploited to mask some inputs directly. The temporal mask ratio in all trials is set to 0.5, which means that only half of the input is retained (details are given in **Part “Evaluating dynamic SNNs on Speck” (page 10)**).

3) Deploy the trained model to Speck. After training the model on GPU (Nvidia RTX 3090), we deploy the trained model, whose weight and threshold have been quantified into INT8 to Speck. Speck greatly eases the difficulty of deploying SNNs on neuromorphic chips by providing a complete toolchain (**page 20, Figure S7, Table S3**). Simply put, use Sinabs-Speck to train your model.

4) Energy assessment. Speck’s power mainly consists of two parts: the DVS camera and the neuromorphic processor. As we stated in **Part “Chip performance evaluation” (page 15)**, **at a supply voltage of 1.2 V, the DVS and processor on Speck typically consume resting power of 0.15 mW and 0.42 mW, respectively.** Since we are using public datasets, the DVS camera with Speck will not be used. Therefore, the power data reported in Table 1 of our paper (page 7) do not include the energy consumption of the DVS camera. We randomly sample 100 samples on each dataset as input to the Speck processor part to evaluate power and latency, where the total power is composed of Logic and RAM power. The duration of each sample is $540ms$. After loading 100 samples, we measured all power consumption and calculated the average and lowest power consumption. For example, **on Gesture, the minimum and average power of dynamic SNN are 0.70mW and 3.8mW (Table 1 of our paper on page 7), respectively.** Power is related to the sparsity of the input event stream. Since Speck is event-driven, fewer events within the same time window (here, the window is $540ms$) mean lower power.

5) Latency assessment. The latency of a single sample is defined as the difference between the timestamp of the output result and the last input event (see **the caption in Table 1 in our paper, page 7**). Since Speck’s input and output are event streams, each event has its own timestamp. We only

need to record the timestamp of the last event in the readout core, and the timestamp of the last event in the input event stream. Speck handles an event stream of 540ms with almost no latency.

Finally, as a summary, we randomly selected 100 samples from each dataset (Gesture/Gait-day/night) and tested the inference power and latency on Speck. The duration of each sample is 540ms. The model deployed on Speck is a network containing three Conv layers and two FC layers. All datasets use the same network structure; the only difference is that the output categories of the last FC (output) layer are different (Gesture is 11 categories, Gait-day/night is 20 categories). Finally, among all 300 samples tested, the lowest inference power of a single sample was 0.70mW, and the inference delay of all samples did not exceed 0.1ms. Note that since we used public datasets, we did not use the camera part of Speck. We declare the 0.42mW as the resting power of the Speck processor part, and the 0.15mW is the resting power consumption of the Speck DVS camera part. Please see Table 3 in this response.

In response, we have made more strict statements about power consumption in all places in the main text, including Abstract, Introduction, and the caption of Table 1, etc.

[1] Amir, Arnon, Brian Taba, David Berg, Timothy Melano, Jeffrey McKinstry, Carmelo Di Nolfo, Tapan Nayak et al. "A low power, fully event-based gesture recognition system." In Proceedings of the IEEE conference on computer vision and pattern recognition, pp. 7243-7252. 2017.

[2] Wang, Yanxiang, Bowen Du, Yiran Shen, Kai Wu, Guangrong Zhao, Jianguo Sun, and Hongkai Wen. "EV-gait: Event-based robust gait recognition using dynamic vision sensors." In Proceedings of the IEEE/CVF Conference on Computer Vision and Pattern Recognition, pp. 6358-6367. 2019.

[3] Wang, Yanxiang, Xian Zhang, Yiran Shen, Bowen Du, Guangrong Zhao, Lizhen Cui, and Hongkai Wen. "Event-stream representation for human gaits identification using deep neural networks." IEEE Transactions on Pattern Analysis and Machine Intelligence 44, no. 7 (2021): 3436-3449.

[4] Yao, Man, Huanhuan Gao, Guangshe Zhao, Dingheng Wang, Yihan Lin, Zhaoxu Yang, and Guoqi Li. "Temporal-wise attention spiking neural networks for event streams classification." In Proceedings of the IEEE/CVF International Conference on Computer Vision, pp. 10221-10230. 2021.

[5] Gallego, Guillermo, Tobi Delbrück, Garrick Orchard, Chiara Bartolozzi, Brian Taba, Andrea Censi, Stefan Leutenegger et al. "Event-based vision: A survey." IEEE transactions on pattern analysis and machine intelligence 44, no. 1 (2020): 154-180.

Comment 2: Furthermore, it is crucial to clarify which components or subsystems of the overall power consumption are included in this measurement. For instance, the author should delineate whether the stated power consumption encompasses the SNN core, memory buffer, dynamic vision sensor (DVS), or any other relevant elements.

Answer to Comment 2: Thanks for the insightful advice. The system architecture of the Speck system is shown in Figure 1. The system consists of the DVS sensor module, the event pre-processing module, a Network on Chip (NoC) module, 9 Spiking CNN (SNN) cores, and a readout core. The DVS pixels respond to the illuminance change in the environment and generate events asynchronously, which go through the pre-processing layer (which can also be bypassed) and then into the SNN cores. The

Figure 1: Speck System Architecture

events are routed between the SNN cores, the pre-processing layer, and the readout core through a NoC. Each SNN core corresponds to one layer of the convolution neural network. Thanks to the event-driven asynchronous computing regime, no core-specific power gating is needed, as the layers (cores) not used by the network will not consume any running power.

For the power consumption evaluation, we use public datasets without using the DVS camera part of Speck to control the consistency of the input data. Therefore, the power estimates in **Table 1 in our paper (page 7)** include the NoC power (data transmission power(events routing)), and the total computational power and memory r/w power that is consumed by the SNN cores, pre-processing block and the readout block.

In response, we added some details in Part “Evaluating dynamic SNNs on Speck” and “Chip performance evaluation”. For example, on page 10, Part “Evaluating dynamic SNNs on Speck”, we added the following statement: *Note, since we use public event-based datasets, the power reported in this Table only involves the processor part of Speck and does not include the DVS camera part of Speck.* On page 15, Part “Chip performance evaluation”, we added the following statement: *In Table 1, we employ three public datasets to test the power of Speck. Since the DVS part of Speck is not exploited, the power in Table 1 is only $P_{\text{processor}}$, including the pre-processing power, the NoC power, the total computational power and memory r/w power that is consumed by the SNN cores, and the readout power.*

Comment 3: *A breakdown of power usage within the system will offer a more precise insight into the chip’s energy-efficient operation and its strengths in handling diverse computational tasks. Could the author kindly provide a power breakdown of the system?*

Answer to Comment 3: Thank you for your suggestion. Power measurements of Speck are not conducted in isolation for individual functional modules but rather based on their actual usage in terms of {DVS pixel analog, DVS pixel digital, processor logic, processor RAM}. Therefore, the chip’s power breakdown can be categorized into four power tracks based on the utilization of its four modules. To give a clearer statement, **we have revised Part “Chip performance evaluation” (page 15).**

Specifically, the measurement result of the resting power of these four power tracks is shown in Table 4. As we stated in **Part “Chip performance evaluation” (page 15)**, the power of Speck is split

Power track	Pixel analog	Pixel digital	Logic	RAM
Power (μW)	71.50	78.07	360	63

Table 4: Resting Power Breakdown of Speck. The power of Speck in the DVS part is: $71.50\mu\text{W} + 78.07\mu\text{W} = 0.149\text{mW} \approx 0.15\text{mW}$. The power of Speck in the processor part is: $360\mu\text{W} + 63\mu\text{W} \approx 0.42\text{mW}$.

into power tracks as is shown:

$$P_{\text{total}} = \underbrace{P_{\text{pixel, analog}} + P_{\text{pixel, digital}}}_{P_{\text{DVS}}} + \underbrace{P_{\text{pre}} + P_{\text{NoC}} + P_{\text{SNN}} + P_{\text{readout}}}_{P_{\text{processor}}}, \quad (1)$$

where P_{DVS} indicates the power of DVS, $P_{\text{processor}}$ denotes the power of the Speck processor. P_{pre} , P_{NoC} , P_{SNN} and P_{readout} correspond to the power of pre-processing core, Network on Chip (NoC), SNN cores and readout core in the processor respectively. **These functional blocks can not be measured separately but can be jointly measured by the P_{Logic} and P_{RAM} depending on their RAM and Logic usage.** The power consumption of the DVS pixels contains the analog part (biases and power of the analog circuits, i.e., $P_{\text{pixel, analog}}$) and the digital part (asynchronous circuits generating and routing out the events, i.e., $P_{\text{pixel, digital}}$). A SynOp which indicates a fundamental neuron computation includes the following steps: logic \rightarrow Kernel RAM \rightarrow logic \rightarrow Neuron RAM Read \rightarrow Neuron RAM Write \rightarrow Logic. The power consumption of the SNN processor is accordingly divided into logic power (all computations for neuron dynamics, i.e., P_{logic}) and RAM power (read/write of kernel and neuron RAMs, i.e., P_{RAM}).

For your convenience, we have reproduced the revised content here.

Chip performance evaluation. We here evaluate Speck’s performance in terms of power and latency. Speck is an event-driven asynchronous chip whose power consumption and output latency vary with the number of SynOp.

Power evaluation. The total power consumption of speck contains four power rails:

$$P_{\text{total}} = \underbrace{P_{\text{pixel, analog}} + P_{\text{pixel, digital}}}_{P_{\text{DVS}}} + \underbrace{P_{\text{pre}} + P_{\text{NoC}} + P_{\text{SNN}} + P_{\text{readout}}}_{P_{\text{processor}}}, \quad (2)$$

where P_{DVS} indicates the power of DVS (see Fig. 5), $P_{\text{processor}}$ denotes the power of the Speck processor. The power consumption of the DVS pixels contains the analog part (biases and power of the analog circuits, i.e., $P_{\text{pixel, analog}}$) and the digital part (asynchronous circuits generating and routing out the events, i.e., $P_{\text{pixel, digital}}$). The power consumption of the processor also includes two parts, P_{Logic} and P_{RAM} . **Due to the split of the power tracks, we exploit the number of SynOps to measure all the power consumption of the processor, this covers all the power consumed by RAM and Logic.** A SynOp includes the following steps: logic \rightarrow Kernel RAM \rightarrow logic \rightarrow Neuron RAM Read \rightarrow Neuron RAM Write \rightarrow Logic. The processor’s power consumption is accordingly divided into logic power (all computations for neuron dynamics, i.e., P_{logic}) and RAM power (read/write of kernel and neuron RAMs, i.e., P_{RAM}). **In summary, Speck’s power breakdown can be categorized into four power tracks based on the utilization of**

its four modules. Power measurements are not conducted in isolation for individual functional modules but rather based on their actual usage in terms of {pixel analog, pixel digital, logic, and RAM}. Thus, P_{Logic} and P_{RAM} can be considered to be the sum of the power consumption of the DVS pre-processing core (P_{pre}), NoC (P_{NoC}), SNN cores (P_{SNN}), and readout core (P_{readout}).

Comment 4: *The author’s assertion of zero running power consumption from logic gates during idle periods is a notable claim that requires further elucidation. To provide a more comprehensive understanding, the author should expound upon the techniques and mechanisms used to accomplish this feat. Specifically, a discussion of power-gating methods, the granularity of power gating, and control mechanisms is essential. Addressing the management of leakage power, including strategies, process technologies, and monitoring, is equally important. Additionally, the author should quantify and discuss the energy and time overhead associated with waking up the chip from its idle state.*

Answer to Comment 4: Thanks for the insightful clarifying question on running (dynamic) power consumption. Since the subject of this work is spike-based dynamic computing, in order to prevent ambiguity about the word “dynamic”, we use “running power” instead of “dynamic power” when evaluating hardware energy consumption. For your question, we have added a lot of details in **Part Materials and Methods - Chip design/Asynchronous logic design methodology**. We explain it as follows.

The asynchronous pipeline design can be viewed as an extreme and most fine-grained form of clock gating. Each component, logic circuitry, pipeline buffers, registers, and SRAM memory only show logic and flip-flop switching on data processing [6,7]. As a clockless design, no pipeline buffers or cache will toggle if nothing is to be processed - our idle state. Using this design in SoC built in its entirety on sparse operation spending its majority idle waiting for data to be processed, the small asynchronous control overhead of having an individual “clock” for every fine-grain pipeline stage is amortized even for less sparse situations. “Gating the clock/control” in an asynchronous controller does not incur any delay between idle and active as it is native to its operation. We do not employ deep sleep power-down techniques, shutting down full pipeline sections of the processing cores to reduce static leakage power consumption. This would be possible with a small overhead of tracking the activity inside a bigger section in our design, to shut down all power. The pipeline design only needs one parallel reset operation of less than a nanosecond after power-up and could be explored in future designs. In Part Material and Methods - Asynchronous logic design methodology, the newly added references [8,9] on the specific type of asynchronous control design used, give the interested reader an extensive overview of the inner workings of this special but well-established design technique.

We tackle the problem of leakage power on an architectural level; by extensively employing on-the-fly computing, we can reduce our required SRAM storage on a chip, reducing both size and leakage power by requiring less memory in the architecture compared to state-of-the-art SNN processors. We point out that one of the reasons for our aforementioned class of asynchronous control structures is to allow for extensive optional voltage scaling to reduce the static and dynamic power consumption on deployed applications. For the results presented in the paper, optional voltage scaling is not active.

We have reproduced the revised Part “Asynchronous logic design methodology” here for your convenience.

Asynchronous logic design methodology. DVS has μ s level temporal resolution due to the asynchronous visual perception paradigm. To complement the architectural advantages of sparse sensing and computing, the neuromorphic chip in Speck is built in a fully asynchronous fashion. The asynchronous data flow architecture provides low latency, and high processing throughput when requested by sensory input, and immediately switches to a low power/idle state when the sensory input is absent. This is achieved by building on the well-established Pre Charge Full Buffer low latency pipeline designs [8]. As each component is naturally gated in this design approach, no complex or slow wake-up procedures must be implemented, thereby reducing running power consumption and obtaining an always-on feature with no additional latency. Asynchronous chips make the data follow event-based timestamps rather than clock rising or falling edges. Therefore, during the idle period, there is no switching output from the DVS pixel array, and there is no information routed to the chip that leads to no running power consumed in any processing unit other than the readout layer. However, there is still a static power consumption from the DVS pixel array, and leak currents from logic and memory, which are reduced by the before mentioned architecture resource optimisations and optional independent voltage scaling for both logic and memory.

In our asynchronous design flow, we implemented an extensive library of asynchronous dataflow templates. Each function in our library is built using a 4-phase handshake and Quasi Delay Insensitive (QDI) Dual Rail (DR) data encoding, making it robust to an extended range of supply voltages, operation conditions and temperatures [7,9]. The main functions implemented in our library are Latch/Buffer, Compose, Splits, Non-conditional Splits, Conditional Pass, and Merge functions [9]. At the last stage of the asynchronous chain, we serialize the data to be monitored to reduce the pin count. Before the serialization operation, we convert the dual rail encoded information to Bundle Data (BD) encoding. Performance is ensured by hierarchically detailed automatic floor planning that employs extensive guides and fences for the individual components and pipeline stages.

[6] Hauck, Scott. "Asynchronous design methodologies: An overview." Proceedings of the IEEE 83, no. 1 (1995): 69-93.

[7] Sparsø, Jens. Introduction to Asynchronous Circuit Design. DTU Compute, Technical University of Denmark, 2020.

[8] Martin, Alain J., Andrew Lines, Rajit Manohar, Mika Nystrom, Paul Penzes, Robert Southworth, Uri Cummings, and Tak Kwan Lee. "The design of an asynchronous MIPS R3000 microprocessor." In Proceedings Seventeenth Conference on Advanced Research in VLSI, pp. 164-181. IEEE, 1997.

[9] Nowick, Steven M., and Montek Singh. "High-performance asynchronous pipelines: An overview." IEEE Design & Test of Computers 28, no. 5 (2011): 8-22.

Comment 5: *In Table 1, it seems that the accuracy loss of Gesture on Speck using dynamic SNN is much more significant (2%) than running on GPU, compared to other cases. Could the author kindly discuss the key points leading to the phenomenon? Also, the author mentions the power and latency is running under 1000 samples. Is the accuracy also obtained from this condition or running under whole datasets?*

Answer to Comment 5: Your questions are very important and have significant practical applications. We argue that the reason for the 2% performance loss may be multi-factorial, involving model deployment, training methods, datasets, algorithm design, and testing methods.

1) Model deployment. Part of the loss comes from the quantization error. on the GPU simulation, most of the computation is carried by float32. On the Speck, the SNN core precision is limited to 8-bit integers for weight values and 16-bit integers for membrane potentials and neuron firing thresholds. When deploying the pre-trained SNNs, all the parameters were quantized into precision range with layer-wise normalizations, this reduced the error but still did not completely eliminate it. For optimization, such as the quantization-aware training can be applied to SNN at the training stage.

2) Training. Part of the accuracy mismatch comes from the alignment between synchronous training and asynchronous processes. Speck operates in a full asynchronous manner which has stochastic properties for individual computing and routing components. The synchronous process, such as GPU, cannot fully simulate this stochastic when using batching and parallelizing techniques. This particular happens when a spiking neuron is about to receive multiple spikes in a short timestep, the Speck core will process each spike exactly follow the spike orders whereas GPU will accumulate the weighted effect into a single timestep. This error can be reduced by sacrificing the acceleration techniques in the training stage, but it is a choice of balance between the training speed and accuracy loss.

3) Datasets. Even if rigorous ablation experiments are used to align all training, deployment, and testing, performance gaps on GPU and Speck may vary for different datasets. In our submission, we provide several demos of Speck applications such as remote controlled car, smart toy, etc. Before implementing these demos, we created datasets in these scenarios and conducted performance tests. We found that on some datasets, there is almost no difference between GPU and Speck accuracy, and some datasets are more sensitive.

4) Algorithm. This is also an interesting finding. We found that different algorithms also affect the performance gap between training and deployment. For example, the training and deployment accuracy gaps of vanilla SNNs and dynamic SNNs are different (see Table 1 of this paper). The performance of the same algorithm on different datasets may also be different. This interesting present was discussed in detail in our previous work [10]. In Figure 7 of [10], the accuracy of the model trained on Gesture will decrease after masking some intermediate features. However, if the same operation is used on the Gait-day dataset, the performance will actually improve.

5) Testing. As stated in Table 1 of this paper, when testing the algorithm on GPU, we used 1000 samples. In contrast, when deploying on Speck, we randomly selected 100 samples. This may cause statistical errors.

Regarding the question “Also, the author mentions the power and latency is running under 1000 samples. Is the accuracy also obtained from this condition or running under whole datasets?”, as we stated in Table 1 of the main text (page 7), and detailed in this Answer, when testing the algorithm on GPU, we used 1000 samples. In contrast, when deploying on Speck, we randomly selected 100 samples. Therefore, the accuracy and energy consumption data obtained by testing the public data set on Speck were obtained on 100 randomly selected samples.

[10] Yao, Man, Hengyu Zhang, Guangshe Zhao, Xiyu Zhang, Dingheng Wang, Gang Cao, and Guoqi Li. ”Sparser spiking activity can be better: Feature refine-and-mask spiking neural network for event-based visual recognition.” Neural Networks 166 (2023): 410-423.

Comment 6: *In Figure 1c, what does network spiking firing rate at each timestep mean? Why does the NSFR of each layer accumulate on the same bar? Could the author kindly*

explain it?

Figure 2: Layer’s Spiking Firing Rate (LSFR) of vanilla SNN. LSFRs of SNNs at each timestep is similar, which indicates that the scales of the activated sub-networks of SNNs are similar for diverse input.

Answer to Comment 6: Thank you for pointing out our oversight. We put the explanation of ”Network Spiking Firing Rate (NSFR)” in **Part “Details of algorithm evaluation (the last paragraph on page 18)”**, but did not indicate it in the caption of Figure 1c.

At the algorithm level, the SNN field often ignores specific hardware implementation details and estimates theoretical energy consumption for a model. This theoretical estimation is just to facilitate the qualitative energy analysis of various SNN algorithms. In general, the energy consumption of SNNs is related to spiking firing, and energy efficiency improves with smaller spike counts. To compute the spike counts, we define four kinds of spiking firing rates.

Assuming that *a single sample* is fed into SNN for inference, there is:

- 1) At timestep t , a spiking network’s **Network Spiking Firing Rate (NSFR)** is the ratio of spikes produced over all the neurons to the total number of neurons in this timestep.
- 2) We define **Network Average Spiking Firing Rate (NASFR)** as the average of NSFR over all timesteps T . NSFR is used to evaluate the change of the same network’s spike distribution at different timesteps; NASFR is exploited to compare the spike distribution of different SNNs.
- 3) Similarly, at timestep t , a **Layer’s Spiking Firing Rate (LSFR)** is the ratio of spikes produced over all the neurons to the total number of neurons in that layer.
- 4) We define the **Layer Average Spiking Firing Rate (LASFR)** that averages LSFR across all timesteps T .

Moreover, for each sample, we will get a set of NSFR, NASFR, LSFR, LASFR. **In this work, by default, we count NSFR, NASFR, LSFR, LASFR of the samples in the entire test set and get their mean as the final data.** Therefore, NSFR in Figure 1c is the result of averaging the NSFR over the entire test set. As for what you mentioned, “Why does the NSFR of each layer accumulate on the same bar?” We just thought, visually, it would be prettier. After your prompting, we realized that this might not be strict, so we replaced the Figure as shown below.

In response, we modified Figure 1c and the corresponding caption.

Comment 7: *It seems that this paper lacks of some details of how to implement the SNN cores. Could the author kindly add some details of implementation?*

Answer to Comment 7: In Part “Material and Methods - Chip Design”, we extended the description of the pipeline processing architecture with a comprehensive step-by-step walkthrough from the sensor over pre-processing, Network on Chip (NoC), SNN processing cores, and readout layer, putting a specific emphasis on the SNN core. Additionally, we present an architecture diagram and the processing pipeline of the SNN cores in the supplementary material Figure S5.

In the paragraph on Part “Asynchronous design methodology”, we reference the pipeline method and control paradigm on which our design is based so that the interested reader can get a very detailed picture of the implementation and design method decisions. We detailed the sensor encoding, added the workings of the NoC design, and described in further detail how a convolution is applied and computed in contrast to classic machine learning CNN implementations.

For your convenience, we copy the design and implementation detail of SNN cores here, along with the newly added Figure S5 (i.e., Figure 3 here).

The event-driven convolution implemented step-by-step in Speck with the following components:

a) Zero padding: the event is padded to retain the layer size if needed. The image field, i.e. the address of the events, is expanded by adding pixels to the borders to retain the image size after the convolution if needed.

b) Kernel anchor and address sweep: In the kernel mapper, the event is first mapped to an anchor point in the output neuron and the kernel space. Using this anchor the kernel, represented by an address, is linked to an address point in the output space. The referenced kernel is swept over the incoming pixel coordinate. The kernel address and the neuron address are swept inversely to each other. For every channel in the output neuron space, the kernel anchor address is incremented so that a new kernel for the new output channel is used. The sweep over the kernel is repeated. If a stride is configured in either the horizontal or vertical direction, the horizontal and vertical sweeps are adjusted to jump over kernel positions accordingly.

c) Address space compression: To effectively use the limited memory space, the verbose kernel address and the neuron address are compressed to avoid unused memory locations. Depending on the configuration, the address space gets packed so that there are no avoidable gaps inside the address that are not used by the configuration.

d) Synaptic kernel memories: The kernel addresses are then distributed on the parallel kernel memory blocks according to the compressed addresses, and the specific signed 8-bit kernel weight is read. The weight and the compressed neuron address are then directed to the parallel neuron compute-in-memory-controller blocks according to the address location. Kernel positions with 0 weight are skipped during reading and are not forwarded to the neuron.

e) Neuron compute units: The compute-in-memory-controller block model a Integrate and Fire (IF) neuron with a linear leak for every signed 16-bit memory word. Besides classic read and write, the memory controller has a read-add-check_spike-write operation. Whenever the accumulated value reaches

Figure 3: The Speck architecture (a) [13]. The left area indicates both the 128×128 event-based vision sensor with its 2D asynchronous readout and the sensor event pre-processing pipeline. The middle area indicates the NoC responsible for all the event routing between all the components. The area indicated on the right incorporates all the nine SNN cores that handle one convolution and one pooling layer each. The SNN cores can optionally be operated as fully connected SNN layers with some restrictions. The small bottom area indicates the decision readout logic. This core enables interfacing to simple synchronous periphery. The convolution core architecture (b) [14]. An event $\{c, x, y\}$ enters the convolution core pipeline, with c as the incoming channel/feature, x as the horizontal coordinate and y as the vertical coordinate. After padding, the event is now expanded to $\{c, x_p, y_p\}$. The Kernel Anchor determines the anchor in both kernel and neuron space $\{c, x_0, y_0, x_0^k, y_0^k\}$. With x_0, y_0 being the anchor in the neuron space and x_0^k, y_0^k for the kernel space. The kernel address sweep now calculates the kernel expansion in x, y and f the output channel/features to $Z * \{(c, f, x^k, y^k), (f, x, y)\}$, with Z being the synaptic fan-out. The parallel address compression packs the storage addresses compact to avoid unused storage gaps for the neuron $(f, x, y) \Rightarrow n_{\text{comp}}$ and kernel $(c, f, x^k, y^k) \Rightarrow k_{\text{comp}}$. Depending on the core, the kernel memory is split into one or multiple memory blocks for parallel access. The kernel value is read from the storage address $k_{\text{comp}}, \{w, n_{\text{comp}}\}$ with w being the signed 8-bit synaptic weight. On a simulation tick, the bias/leak sweep will generate a pair of $\{b_{\text{comp}}, n_{\text{comp}}\}$ for every active neuron, the address b_{comp} gets read in the bias/leak memory and forwarded as $\{w, n_{\text{comp}}\}$ with the kernel events to the neuron. Depending on the core, the neuron unit is split into one or multiple parallel compute units. The address space decompression turns the $\{n_{\text{comp}}\}$ back to $\{f, x, y\}$. The sum pooling operates on the same event structure $\{f, x_s, y_s\}$. And the Channel shift and routing prepare it for routing $S * \{d_x, f_s, x_s, y_s\}$ with S being the source fan-out of 1 or 2, d_x corresponding to the destination id and f_s being the arithmetically shifted destination channel. The neuron compute unit (c) [14]. It uses in-memory-controller compute to model the LIF neuron model. The flow control at the input ensures that the controller always has a bubble and is therefore deadlock-free. The signed 16-bit neuron state variable gets read, modified by the signed synaptic or bias input, compared and written back. In case a threshold condition is met, the $\{n_{\text{comp}}\}$ is sent out to indicate the corresponding neuron spiked. The above-threshold condition can both trigger a subtraction operation or a reset to a fixed value of the corresponding neuron state variable. The state variable cannot cross a configured lower bound and will be clamped to that value in case any operation brings the variable below it.

a configured threshold, an event is sent out and the neuron state variable has a threshold subtraction or reset written back.

f) Bias and leak address sweep and memory: The leak (or bias) is modelled via an additional memory controller. The Leak/bias controller has a neuron individual signed 16-bit weight stored for every output channel map. An update event with this bias is sent to all its active neurons on a time reference tick. The reference tick is supplied from off-chip and is fully user-configurable.

g) Pooling: The output events are finally merged into a pooling stage. The pooling stage operates on the sum pooling principle, i.e., it merges the events from 1,2 or 4 neurons in both x and y coordinates individually.

h) Channel shift and routing: Before entering the routing NoC, the channels are shifted, and a prefix with routing information is added. One event is sent per destination for up to 2.

Speck has 9 SNN cores (layers) with different computational capabilities and memory sizing. For example, SNN core0, SNN core1, and SNN core2 have larger neuron memory sizes because the first layer, which connects to the dynamic vision sensor part, requires more neuron states with fewer input channels. As the network gets deeper, the synaptic operation or kernel memory requirement increases. The intermediate cores usually require a larger kernel memory size for generating more output channels using different kernel filters.

A key point for our presented architecture is the synaptic memory utilisation. Especially for CNN-based architectures, the on-the-fly computation of synaptic connections allows for minimising memory requirements. This, in turn, saves area and energy - in the case of SRAM, both running and static. Our dedicated sCNN approach allows for many more synaptic connections by using the kernel weights stored in memory and computing all the synapses that share weights compared to standard SNN hardware implementations [11,12] with minimal additional compute required. As shown in Table S1, on-the-fly synaptic kernel mapping allows deployment of bigger network models to larger feature size CMOS, thus significantly more cost-effective fabrication technologies while matching state-of-the-art performance. Besides exploiting SNN cores as convolutional layers, any SNN core that can be utilised as a fully connected layer with synaptic connections up to 64k, 32k, or 16k. In general, this is preferred at the last stages of the SNN.

[11] Merolla, Paul A., John V. Arthur, Rodrigo Alvarez-Icaza, Andrew S. Cassidy, Jun Sawada, Filipp Akopyan, Bryan L. Jackson et al. "A million spiking-neuron integrated circuit with a scalable communication network and interface." *Science* 345, no. 6197 (2014): 668-673.

[12] Davies, Mike, Narayan Srinivasa, Tsung-Han Lin, Gautham Chinya, Yongqiang Cao, Sri Harsha Choday, Georgios Dimou et al. "Loihi: A neuromorphic manycore processor with on-chip learning." *IEEE Micro* 38, no. 1 (2018): 82-99.

[13] Tugba Demirci, Q. N., Sheik Sadique & Ole, R. Event-driven integrated circuit having interface system (2022). WIPO Patent App No. PCT/CN2021/088143.IPN.WO2022/221994A1.

[14] Richer Ole, L. Q., Qiao Ning & Sadique, S. Event-driven spiking convolutional neural network (2020). WIPO Patent App No. PCT/EP2020/059798.IPN.WO2020/207982A1.

3 Response to Reviewer #3

The document details the development of “Speck”, an innovative neuromorphic chip, emphasizing its low-power consumption and dynamic computing abilities. The neuromorphic chip “Speck” showcases a novel design approach, potentially advancing the field of neuromorphic computing. The paper provides detailed technical descriptions of the chip’s architecture and functions. The work contributes to the ongoing research in low-power, brain-inspired computing, which is a significant and growing area of interest in the field of computer science.

General Response to Reviewer #3. We are very pleased that you considers our work “the paper presents an interesting and potentially significant advancement in neuromorphic computing”. We greatly appreciate you for spending valuable time in providing insightful comments and suggestions on the manuscript. This truly helped us to improve it. We have considered your questions very carefully and have responded point-to-point. In addition, we have revised the main text accordingly to be consistent with the response letter. We hope that you will find our responses satisfactory. We will be happy to supply any technical details you require.

Comment 1: *The description of the neuromorphic chip’s design and function could be more detailed. It’s unclear how certain design choices impact the chip’s performance, which might lead to questions about the reproducibility and scalability of the results.*

Answer to Comment 1: Thank you for your focused question for further clarification. Based on your comments, we’ve given more details about the architecture implementation and how the spiking CNN (sCNN) method is manifested in hardware. We comprehensively extended the **Part “Materials and Methods - Chip Design”** and added a **Figure**.

Specifically, the description of the pipeline processing architecture with a step-by-step walkthrough from the sensor over pre-processing, Network on Chip, SNN processing cores and readout layer, putting a specific emphasis on the SNN core for better understanding. We added a high-level chip architecture and a detailed processing pipeline diagram of the SNN cores in the **Figure S5**. The supplementary material focuses on the data transformations throughout the SNN pipeline, giving the reader an in-depth idea of how the spiking CNN algorithm is transformed into a hardware processing pipeline. This will also allow the reader to discover the synergies between the sparse algorithm, sparse sensor and the truly event-driven processing pipeline architecture. We pointed out the scaling of parallelism inside the pipeline matching with the increasing sparsity of the sCNN method for increasing layers.

In the **paragraph Asynchronous design methodology** we detail the design method choices and point out that one significant reason for this specific pipeline controller is the low latency and robustness over an extended range of operation temperatures and robustness to voltage scaling for further power reduction. We point out why the Asynchronous design method excels in dynamic power reduction by only consuming power on demand, used in an application-driven design optimised for sparse operation.

For your convenience, we have reproduced the revised content here.

Chip design. We define Speck as an efficient medium-scale neuromorphic sensing-computing edge hardware. It has four main components: DVS with a spatial resolution of 128×128 , sensor pre-processing core, 9-SNN cores enabling combined convolution and pooling or fully connected SNN layers, and readout core. The connection of each component mentioned is done through a universal event router. Combining such low latency, high dynamic range and sparse sensor with an event-driven spiking Convolutional Neural Network (sCNN) processor, that excels in real-time low latency processing on a single SoC is a natural technological step. To complement the architectural advantages of always-on sparse sensing and computation, the SoC is built in a fully asynchronous fashion. The asynchronous data flow architecture provides low latency and high throughput processing when requested by sensory input while inherently shifting to a low power/idle state when the sensory input is absent. Thoroughly investigating and verifying many edge computing vision application scenarios, Speck was designed to comprise 328K neuron units spread over nine high-speed, low latency and low power SNN cores.

Sensing-computing coupling. Current technologies exploit USB connections or other interface technologies to connect to the vision sensor and neuromorphic chips/processors. Moving data over chip boundaries and long distances impacts latency and increases power dissipation for robust signal transmission significantly. While advanced CIS-CMOS processes can couple dedicated high-quality vision sensors (vision optimised fabrication process) and neural network compute chips (logic optimised fabrication process) in a single package, by combining both sensor and processing on a single die into a smart sensor, we lower unit production costs significantly while saving energy on high-speed and low-latency data communication, as the raw sensory data never has to leave the chip.

The Sensor. The sensor of Speck consists of 128×128 individually operating event-based vision pixels, also called dynamic vision pixels. In contrast to the frame-based cameras, these pixels encode the incident light intensity temporally on a logarithmic intensity scale, also known as Temporal Coding (TC) encoding. In other words, the sensor transmits only novel information in the field of view, sparsifying the data stream significantly and seizing transmission on no visual changes. Each pixel is attached to a single handshake buffer to enable the pixel to work fully independently from the arbitration readout system. The arbitration is built out of one arbiter tree for column arbitration and one for the rows. The event address is encoded from the acknowledge signals of the arbitration trees and handed off as an Address Event Representation (AER) word to the event pre-processing block. The events are encoded as 1-bit polarity (ON-increasing illumination / OFF-decreasing illumination), 7-bit x -address, and 7-bit y -address (total of 15-bit data per pixel event). A complete arbitration process with ID encoding takes approximately $2.5 - 7.5ns$ for a single readout. The arbitration endpoint with buffer in the pixel itself is optimised to limit the transistor count, resulting in a fill factor of 45% front illumination for each pixel. The refractory of each pixel is around $500 \mu s$. Under the worst-case condition where the activity rate of the image is around 100%, the data needs to be transmitted during refractory is around 250 Kb ($128 \times 128 \times 15bit$), i.e. 500 Mb/s data rate. For a typical condition, 10% to 20% of the pixel area is estimated to be active, which causes a 50 to 100Mb/s data rate. Opposed to power-hungry high-speed low-latency off-chip communication, transmitting this sparse data stream on-chip yields a significant reduction in power consumption, proportional to the data rate.

DVS pre-processing core. To conform the raw AER event stream from the sensor to the requirements of the sCNN, a pre-processing stage is required. The image may be flipped, rotated or cropped if only a Region of Interest (ROI) of the image is required. A lower image resolution might be required, or the polarity can be ignored. To accomplish this, the sensor event pre-processing pipeline consists of multiple stages enabling the following adjustments in the sensor event stream: polarity selection as ON only, OFF only, or both, region of interest adjustment, image mirroring in x , y or both axes, pooling in

x or y coordinates separately or together, shifting the origin of the image to another location by adding an offset, etc. In addition, there is also an option to filter out the noisy events coming out from the DVS by using full-digital low-pass or high-pass filters. The output of the pre-processing core has a maximum of two channels, with each channel indicating whether the event belongs to the ON or OFF category. After the pre-processing of the sensory events, the data output is transmitted to the Network on Chip (NoC).

Network on Chip. The NoC router follows a star topology. The routing system operates non-blocking for any feed-forward network model and routes events via AER connections. The mapping system allows data to be sent from one convolution core to up to 2 other cores and for one core to receive events from multiple sources without addressing superposition with up to 1024 incoming feature channels. On every incoming channel, the routing header of every AER packet is read, and the payload is directed to the destination. This is done by establishing parallel physical routing channels that do not intersect for any network topology that does not contain recurrence. This prevents skew due to other connections and deadlocks by loops inside the pipeline structure. The routing header information is stripped from the word during transport, and the payload is delivered to its intended destination.

SNN core. In conventional CNN, a frame-based convolution is done. Thus, the camera’s full frame must be available before starting the convolution operation. In contrast to CNN, event-driven sCNN does not operate on a full frame basis: for every arriving pixel event, the convolution is computed for only that pixel position. For a given input pixel, all output neurons that are associated with its convolution are traversed, as opposed to a kernel that is swept pixel-by-pixel over a complete image. An incoming event includes the x and y coordinates of the active pixel as well as the input channel c it belongs to. The event-driven convolution implemented step-by-step in Speck with the following components (Figure S5, i.e., Figure 4 here). The event-driven convolution implemented step-by-step in Speck with the following components:

a) Zero padding: the event is padded to retain the layer size if needed. The image field, i.e. the address of the events, is expanded by adding pixels to the borders to retain the image size after the convolution if needed.

b) Kernel anchor and address sweep: In the kernel mapper, the event is first mapped to an anchor point in the output neuron and the kernel space. Using this anchor the kernel, represented by an address, is linked to an address point in the output space. The referenced kernel is swept over the incoming pixel coordinate. The kernel address and the neuron address are swept inversely to each other. For every channel in the output neuron space, the kernel anchor address is incremented so that a new kernel for the new output channel is used. The sweep over the kernel is repeated. If a stride is configured in either the horizontal or vertical direction, the horizontal and vertical sweeps are adjusted to jump over kernel positions accordingly.

c) Address space compression: To effectively use the limited memory space, the verbose kernel address and the neuron address are compressed to avoid unused memory locations. Depending on the configuration, the address space gets packed so that there are no avoidable gaps inside the address that are not used by the configuration.

d) Synaptic kernel memories: The kernel addresses are then distributed on the parallel kernel memory blocks according to the compressed addresses, and the specific signed 8-bit kernel weight is read. The weight and the compressed neuron address are then directed to the parallel neuron compute-in-memory-controller blocks according to the address location. Kernel positions with 0 weight are skipped during reading and are not forwarded to the neuron.

Figure 4: The Speck architecture (a) [1]. The left area indicates both the 128×128 event-based vision sensor with its 2D asynchronous readout and the sensor event pre-processing pipeline. The middle area indicates the NoC responsible for all the event routing between all the components. The area indicated on the right incorporates all the nine SNN cores that handle one convolution and one pooling layer each. The SNN cores can optionally be operated as fully connected SNN layers with some restrictions. The small bottom area indicates the decision readout logic. This core enables interfacing to simple synchronous periphery. The convolution core architecture (b) [2]. An event $\{c, x, y\}$ enters the convolution core pipeline, with c as the incoming channel/feature, x as the horizontal coordinate and y as the vertical coordinate. After padding, the event is now expanded to $\{c, x_p, y_p\}$. The Kernel Anchor determines the anchor in both kernel and neuron space $\{c, x_0, y_0, x_0^k, y_0^k\}$. With x_0, y_0 being the anchor in the neuron space and x_0^k, y_0^k for the kernel space. The kernel address sweep now calculates the kernel expansion in x, y and f the output channel/features to $Z * \{(c, f, x^k, y^k), (f, x, y)\}$, with Z being the synaptic fan-out. The parallel address compression packs the storage addresses compact to avoid unused storage gaps for the neuron $(f, x, y) \Rightarrow n_{\text{comp}}$ and kernel $(c, f, x^k, y^k) \Rightarrow k_{\text{comp}}$. Depending on the core, the kernel memory is split into one or multiple memory blocks for parallel access. The kernel value is read from the storage address $k_{\text{comp}}, \{w, n_{\text{comp}}\}$ with w being the signed 8-bit synaptic weight. On a simulation tick, the bias/leak sweep will generate a pair of $\{b_{\text{comp}}, n_{\text{comp}}\}$ for every active neuron, the address b_{comp} gets read in the bias/leak memory and forwarded as $\{w, n_{\text{comp}}\}$ with the kernel events to the neuron. Depending on the core, the neuron unit is split into one or multiple parallel compute units. The address space decompression turns the $\{n_{\text{comp}}\}$ back to $\{f, x, y\}$. The sum pooling operates on the same event structure $\{f, x_s, y_s\}$. And the Channel shift and routing prepare it for routing $S * \{d_x, f_s, x_s, y_s\}$ with S being the source fan-out of 1 or 2, d_x corresponding to the destination id and f_s being the arithmetically shifted destination channel. The neuron compute unit (c) [2]. It uses in-memory-controller compute to model the LIF neuron-model. The flow control at the input ensures that the controller always has a bubble and is therefore deadlock-free. The signed 16-bit neuron state variable gets read, modified by the signed synaptic or bias input, compared and written back. In case a threshold condition is met, the $\{n_{\text{comp}}\}$ is sent out to indicate the corresponding neuron spiked. The above-threshold condition can both trigger a subtraction operation or a reset to a fixed value of the corresponding neuron state variable. The state variable cannot cross a configured lower bound and will be clamped to that value in case any operation brings the variable below it.

e) Neuron compute units: The compute-in-memory-controller block models an Integrate and Fire (IF) neuron with a linear leak for every signed 16-bit memory word. Besides classic read and write, the memory controller has a read-add-check_spike-write operation. Whenever the accumulated value reaches a configured threshold, an event is sent out and the neuron state variable has a threshold subtraction or reset written back.

f) Bias and leak address sweep and memory: The leak (or bias) is modelled via an additional memory controller. The Leak/bias controller has a neuron individual signed 16-bit weight stored for every output channel map. An update event with this bias is sent to all its active neurons on a time reference tick. The reference tick is supplied from off-chip and is fully user-configurable.

g) Pooling: The output events are finally merged into a pooling stage. The pooling stage operates on the sum pooling principle, i.e., it merges the events from 1, 2 or 4 neurons in both x and y coordinates individually.

h) Channel shift and routing: Before entering the routing NoC, the channels are shifted, and a prefix with routing information is added. One event is sent per destination for up to 2.

Speck has 9 SNN cores (layers) with different computational capabilities and memory sizing. For example, SNN core0, SNN core1, and SNN core2 have larger neuron memory sizes because the first layer, which connects to the dynamic vision sensor part, requires more neuron states with fewer input channels. As the network gets deeper, the synaptic operation or kernel memory requirement increases. The intermediate cores usually require a larger kernel memory size for generating more output channels using different kernel filters.

A key point for our presented architecture is the synaptic memory utilisation. Especially for CNN-based architectures, the on-the-fly computation of synaptic connections allows for minimising memory requirements. This, in turn, saves area and energy - in the case of SRAM, both running and static. Our dedicated sCNN approach allows for many more synaptic connections by using the kernel weights stored in memory and computing all the synapses that share weights compared to standard SNN hardware implementations [11,12] with minimal additional compute required. As shown in Table S1, on-the-fly synaptic kernel mapping allows deployment of bigger network models to larger feature size CMOS, thus significantly more cost-effective fabrication technologies while matching state-of-the-art performance. Besides exploiting SNN cores as convolutional layers, any SNN core that can be utilised as a fully connected layer with synaptic connections up to 64k, 32k, or 16k. In general, this is preferred at the last stages of the SNN.

Readout core. The optimal readout engine of the Speck is essential to receive the classification output directly from the chip. The last core output of the SNN can be connected to the readout engine. Unlike the neuromorphic chip's other cores (layers), only one SNN core can be connected to the readout layer for a given network configuration. The readout layer can simultaneously calculate 15 different classifications connected to 15 different output channels of the last SNN core. Each channel has a parallel processing engine that calculates the average of spike counts over 1, 16, or 32 slow clock cycles, in the range of 1kHz to 50kHz operation. Furthermore, an optional asynchronous internal clock is also generated by the event activity of the DVS and can be used as a timing tick of the spike count averaging function. After computing the average of each spiking channel, the average value is compared by a global threshold that is the same for all 15 readout engines. The average values that exceed this threshold are compared, and the one with the maximum value is selected as the classification winner. The index of the winner neuron or spiking channel is directly sent to the readout pins. When a network requires more than 15 classifications, the readout layer can be bypassed or not used, and the spike information of up to 1024

output feature maps can be read from the last SNN core output. To get a reasonable identification, an external processor is required to do the averaging over time and find the right class selection.

Asynchronous logic design methodology. DVS has μ s level temporal resolution due to the asynchronous visual perception paradigm. To complement the architectural advantages of sparse sensing and computing, the neuromorphic chip in Speck is built in a fully asynchronous fashion. The asynchronous data flow architecture provides low latency, and high processing throughput when requested by sensory input, and immediately switches to a low power/idle state when the sensory input is absent. This is achieved by building on the well-established Pre Charge Full Buffer low latency pipeline designs. As each component is naturally gated in this design approach, no complex or slow wake-up procedures must be implemented, thereby reducing running power consumption and obtaining an always-on feature with no additional latency. Asynchronous chips make the data follow event-based timestamps rather than clock rising or falling edges. Therefore, during the idle period, there is no switching output from the DVS pixel array, and there is no information routed to the chip that leads to no running power consumed in any processing unit other than the readout layer. However, there is still a static power consumption from the DVS pixel array, and leak currents from logic and memory, which are reduced by the before mentioned architecture resource optimisations and optional independent voltage scaling for both logic and memory.

In our asynchronous design flow, we implemented an extensive library of asynchronous dataflow templates. Each function in our library is built using a 4-phase handshake and Quasi Delay Insensitive (QDI) Dual Rail (DR) data encoding, making it robust to an extended range of supply voltages, operation conditions and temperatures. The main functions implemented in our library are Latch/Buffer, Compose, Splits, Non-conditional Splits, Conditional Pass, and Merge functions. At the last stage of the asynchronous chain, we serialize the data to be monitored to reduce the pin count. Before the serialization operation, we convert the dual rail encoded information to Bundle Data (BD) encoding. Performance is ensured by hierarchically detailed automatic floor planning that employs extensive guides and fences for the individual components and pipeline stages.

[1] Tugba Demirci, Q. N., Sheik Sadique & Ole, R. Event-driven integrated circuit having interface system (2022). WIPO Patent App No. PCT/CN2021/088143.IPN.WO2022/221994A1.

[2] Richer Ole, L. Q., Qiao Ning & Sadique, S. Event-driven spiking convolutional neural network (2020). WIPO Patent App No. PCT/EP2020/059798.IPN.WO2020/207982A1.

Comment 2: *The paper could benefit from a more robust experimental section. Specifically, comparisons with other neuromorphic chips or standard computing systems would provide a clearer context for the chip’s performance and efficiency.*

Answer to Comment 2: Many thanks for the careful review. In this response, we first systematically provide more details about Speck’s power consumption metrics and testing procedures. Relevant details are given in Part “Materials and Methods - Chip performance evaluation” in the main text.

Chip performance evaluation. We evaluate Speck’s performance in terms of power and latency. Speck is an event-driven asynchronous chip whose power consumption and output latency vary with the number of SynOp.

Figure 5: DVS power consumption measurement based on the linear relationship between event rate and power.

	Resting Power (mW)	Running Power (pJ/SynOp)
Logic	0.36	444 ± 2.7
RAM	0.06	177 ± 1.6

Table 5: **Power consumption of Speck with 9 SNN cores.** The resting power is about 0.42mW. The data in this table are obtained from Figure 6.

Power evaluation. The total power consumption of speck contains four power rails:

$$P_{\text{total}} = \underbrace{P_{\text{pixel, analog}} + P_{\text{pixel, digital}}}_{P_{\text{DVS}}} + \underbrace{P_{\text{pre}} + P_{\text{NoC}} + P_{\text{SNN}} + P_{\text{readout}}}_{P_{\text{processor}}} + \overbrace{P_{\text{Logic}} + P_{\text{RAM}}} \quad (3)$$

where P_{DVS} indicates the power of DVS (see Fig. 5), $P_{\text{processor}}$ denotes the power of the Speck processor. The power consumption of the DVS pixels contains the analog part (biases and power of the analog circuits, i.e., $P_{\text{pixel, analog}}$) and the digital part (asynchronous circuits generating and routing out the events, i.e., $P_{\text{pixel, digital}}$). The power consumption of the processor also includes two parts, P_{Logic} and P_{RAM} . Due to the split of the power tracks, we exploit the number of SynOps to measure all the power consumption of the processor, this covers all the power consumed by RAM and Logic. A SynOp includes the following steps: logic \rightarrow Kernel RAM \rightarrow logic \rightarrow Neuron RAM Read \rightarrow Neuron RAM Write \rightarrow Logic. The processor’s power consumption is accordingly divided into logic power (all computations for neuron dynamics, i.e., P_{logic}) and RAM power (read/write of kernel and neuron RAMs, i.e., P_{RAM}). In summary, Speck’s power breakdown can be categorized into four power tracks based on the utilization of its four modules. Power measurements are not conducted in isolation for individual functional modules but rather based on their actual usage in terms of {pixel analog, pixel digital, logic, and RAM}. Thus, P_{Logic} and P_{RAM} can be considered to be the sum of the power consumption of the DVS pre-processing core (P_{pre}), NoC (P_{NoC}), SNN cores (P_{SNN}), and readout core (P_{readout}).

For each power rail, the energy consumption can be divided into resting and running power, which could be measured separately in the following way:

Figure 6: **Unit power measurement based on the linear relationship between SynOps/s and power.** Synaptic Operations (SynOp) is the basic unit of energy consumption assessment in Speck, which is defined as all the steps involved in the life-cycle of a spike arriving at a layer until it updates the neuron states and generates a spike if applied. A SynOp includes the following steps: logic \rightarrow Kernel RAM \rightarrow logic \rightarrow Neuron RAM Read \rightarrow Neuron RAM Write \rightarrow Logic. Thus, whenever a spike arrives at a core, the power consumption can be roughly divided into two parts: RAM power and Logic power. We send random events with fixed firing rates to a specific layer to trigger computations, and measure the RAM and Logic power of the chip. Then, we get a curve of power-SynOps, and the results are shown in the figure. The intercept and slope of the fitted straight line represent resting power and energy consumption of a single SynOp, respectively.

	Input	Output	Kernal	Stride
Conv-1	2	32	3	1
Conv-2	32	32	3	2
Conv-3	32	32	3	1
AvgPool			4	4
FC-1	512	32		
FC-2	32	11		

Table 6: Network structure for testing latency

1. Design a list of stimuli which will induce known events/SynOps rate r_1, r_2, \dots at the circuit under test (e.g. flickering light with fixed frequency for DVS, pseudo-random input spike stream for SNN processor, etc.)
2. Measure the average power consumption P_1, P_2, \dots over time for each stimulus
3. Fit a straight line to $\mathbf{P} = P_{\text{rest}} + \mathbf{r}E_{\text{run}}$. The estimated resting power is be P_{rest} and the running energy per spike/operation is E_{run} .

The results on Speck are shown in Figure 6 (i.e., Figure S6 in the paper) and Table 5 (i.e., Table S2 in the paper). The spiking firing rate can effectively affect power consumption with respect to event-driven processing. **At a supply voltage of 1.2 V, the DVS and SNN cores on Speck typically consume resting power consumption of 0.15 mW and 0.42 mW, respectively.**

Table 7: Comparison of the Speck chip with existing neuromorphic chips. Speck is the first sensing-computing neuromorphic SoC in the world, which is defined as an efficient medium-scale edge computing hardware that can meet the needs of a variety of edge visual scenarios in terms of high accuracy, low power, and low latency. Unlike other neuromorphic chips, Speck integrates a DVS with μ s level temporal resolution to perceive visual information sparsely. Benefiting from fully asynchronous logic design, Speck has extremely low rest power consumption, thus realizing the always-on profile in edge computing scenes. In contrast, classic neuromorphic chips, such as TrueNorth and Loihi, generally use a partially asynchronous design, i.e. globally asynchronous locally synchronous, or globally synchronous locally asynchronous. Compared to the earlier asynchronous neuromorphic chip Neurogrid, which used mixed-analog-digital circuits, Speck exploits a fully asynchronous digital design method that is suitable for the design of large-scale SNNs. In addition, the unique design of asynchronous convolution in Speck makes its running power consumption very low. It consumes only 0.42-15mW in typical vision application scenarios, normally only few mWs for most classical scenario applications (Normalized to a 65-nm CMOS node, and optionally to a 1.2-V supply voltage.)

Platform	BrainScales	SpiNNaker	Neurogrid	TrueNorth	Darwin	Loihi	Loihi-2	Tianjic	Speck
Model	SNN	SNN	SNN	SNN	SNN	SNN	SNN	ANN/ SNN	SNN
Power	1300mW	1000mW @180MHz	150mW	63-300mW	58.8mW @1.8V+70MHz	74mW	N. A.	950mW@1.2V 400mW@0.9V	0.42-15mW @1.2V
Clock	Partially Async	Partially Async	Async	Partially Async	Sync	Partially Async	Partially Async	Sync	Async
Implementation Technique	Mixed- Analog-Digital	Digital	Mixed- Analog-Digital	Digital	Digital	Digital	Digital	Digital	Digital
Vision Sensor	No	No	No	No	No	No	No	No	Yes
Technology	180nm	130nm	180nm	28nm	180nm	14nm	7nm	28nm	65nm
Area (mm ²)	50	102	170	430	25	60	31	14.44	30
Number of Neuron	512	18k	65k	1M	2048	131k	1M	40k	328k
Number of Synapses	128k	18M	100M	256M	4M	128M	120M (> for CNN)	9.75M	279k (up to 6.16G for CNN)
Normalized Neuron Density (Neu/mm ²)	79	706	2.93k	431	628	1011	374	514	11k
Normalized Synaptic Density (Syn/mm ²)	20k	705k	4.51M	110k	1.23M	99k	45k	125k	9.3k
Normalized Power (process only) (mW/mm ²)	3600	2000	415	27	163	16	-	172	0.42
Normalized Power (process & V _{DD}) (mW/mm ²)	-	2000	2596	11	366	6.2	-	97	0.42

Latency evaluation. We measure the latency of the Speck by calculating the difference between an input event and an output event. Speck can be flexibly configured to include different modules in the pipeline, including the DVS pixels, pre-processing, and different SNN layers.

1. DVS response latency, defined as the time difference between the change of light intensity and the generation of the corresponding event, is measured to range from 40 μ s to 3 ms, depending on the bandwidth configuration.
2. DVS pre-processing layer latency, defined as the time to adapt the raw DVS events to SNN input spikes, including pooling, ROI (Region of Interest selection, mirroring, transposition and multicasting, is measured to be 40 ns.
3. SNN processor latency (per layer), defined as the time it takes to perform the event-driven convolution. It is measured by configuring a kernel with ones in a stride \times stride square, and threshold=1 (thus guarantees exactly one output spike is generated for every input spike). The value is related

to the kernel size and the relative position of the neuron in the kernel, and ranges from 120 ns to 7 μ s (per layer).

4. IO delay: Speck uses a customised serial interface for event input and output. The transmission time is 32 and 26 IO interface clock cycles for spike input and output, respectively. The data are converted to/from parallel lines inside the chip for fast intra-chip transmission and processing. The conversion time, including both input and output, is 125 ns.

To evaluate the end-to-end latency of the SNN processor on Speck, we map the network in Table 6 onto the chip and record the input and output spikes. Since the chip is fully asynchronous and feed-forward, an output spike only takes place after the corresponding input spike triggers at least one spike in each layer sequentially. We measure the minimum delay between any input-output spike-pair to be 3.36 μ s (averaged over all samples). This includes the input spike transmission time of 1.28 μ s at 25 MHz IO interface clock.

Then, we compare Speck side-by-side with more similar chips (based on published public data). The results of the comparison are given in Table S1. We have reproduced it here for your convenience, i.e., Table 7. Unlike other neuromorphic chips, Speck integrates a DVS with μ s level temporal resolution to perceive visual information sparsely. Benefiting from fully asynchronous logic design, Speck has extremely low rest power consumption, thus realizing the always-on profile in edge computing scenes. In contrast, classic neuromorphic chips, such as TrueNorth and Loihi, generally use a partially asynchronous design, i.e. globally asynchronous locally synchronous, or globally synchronous locally asynchronous. The unique design of asynchronous convolution in Speck makes its running power consumption very low. It consumes only 0.42-15mW in typical vision application scenarios, normally only few mWs for most classical scenario applications (Normalized to a 65-nm CMOS node, and optionally to a 1.2-V supply voltage.) It can be seen that Speck has obvious advantages of low power consumption in edge vision scenarios.

Comment 3: *The paper sometimes generalizes its findings without acknowledging the specific context or limitations of the study. This could lead to an overestimation of the chip’s applicability in diverse computing environments. For instance, it suggests that the chip’s efficiency and low-power consumption could revolutionize how computing tasks are performed, especially in energy-constrained environments.*

Overall, while the paper presents an interesting and potentially significant advancement in neuromorphic computing, strengthening these areas could enhance its scientific rigor and credibility. However, the paper could benefit from a more structured discussion section that directly addresses potential limitations and broader implications of the research.

Answer to Comment 3: Thank you for your insightful review. As presented in our case, asynchronous pipeline design matches impeccably well with the processing requirement of sparse SNN algorithms. Speck’s sCNN architecture is very application-specific and gains a lot of optimisation from limiting the network type and scopes; while other general-purpose SNN architectures are not as efficient in power and memory usage, they enable a much broader class of network types with the penalty of using much more ASIC area. In the **Discussion**, touching on the limitations of Speck, we try to give the reader a pointer to the restrictions imposed by the neuromorphic-vision-specific edge computing hardware design.

For your convenience, we have reproduced the revised content here.

At the hardware level, we have demonstrated Speck, which bringing to reality the theoretical advantages of dynamic SNNs at the algorithmic level. The most intriguing feature of Speck is the extremely low resting power (no-input consumes no running energy) brought about by the fully asynchronous design, i.e., always-on, making it particularly competitive in resource-constrained edge computing scenarios. This is also the basic hardware requirement for dynamic computing. We have demonstrated that the energy gain from the sophisticated dynamic algorithm design is completely negligible once the resting power is too high. Moreover, present-day neuromorphic computing frequently separates the design of applications, algorithms, and chips. The needs of hardware and applications are rarely considered when designing neuromorphic algorithms and vice versa. By contrast, Speck incorporates a fully asynchronous spike-based neuromorphic chip with a DVS camera, creating the perfect blend of hardware and applications well suited for dynamic computing. Calculations in Speck are only triggered when DVS generates an event. Comprehensively, based on our top-level design of dynamic algorithms, chip architecture, and real-world application requirements, we have demonstrated mW-level power and ms-level latency solution in typical dynamic visual scenarios. This tapping into the potential of neuromorphic computing will undoubtedly advance the field.

In target edge computing environments, the overhead energy is strictly constrained, especially for a small system working in self-powered mode for a long time. Speck is the first neuromorphic chip with sensing-computing-integrated functionality, which consumes quite low power consumption via asynchronous digital design. However, such high energy efficiency and low production are difficult to model with flexibility and computing precision. Fortunately, it is acceptable in our target scenarios where energy efficiency matters more than the task difficulty and behavior accuracy. We believe Speck can cover a broad range of neuromorphic-vision-specific edge computing tasks distinct from cloud computing, while improving modeling flexibility and computing precision under the energy constraint remains an interesting and valuable direction. For example, enriching the supported network types and introducing mixed-precision computing might be possible solutions in future work.

REVIEWERS' COMMENTS

Reviewer #1 (Remarks to the Author):

The revised manuscript has answered my comment. I have no further comment

Reviewer #3 (Remarks to the Author):

The authors addressed all comments to my satisfaction. I recommend including some of the responses in any supplementary material to the article.